# FAST SAMPLING OF DIFFUSION MODELS WITH EXPONENTIAL INTEGRATOR

**Qinsheng Zhang**
Georgia Institute of Technology
qzhang419@gatech.edu

**Yongxin Chen**
Georgia Institute of Technology
yongchen@gatech.edu

## ABSTRACT

The past few years have witnessed the great success of Diffusion models (DMs) in generating high-fidelity samples in generative modeling tasks. A major limitation of the DM is its notoriously slow sampling procedure which normally requires hundreds to thousands of time discretization steps of the learned diffusion process to reach the desired accuracy. Our goal is to develop a fast sampling method for DMs with fewer steps while retaining high sample quality. To this end, we systematically analyze the sampling procedure in DMs and identify key factors that affect the sample quality, among which the method of discretization is most crucial. By carefully examining the learned diffusion process, we propose Diffusion Exponential Integrator Sampler (DEIS). It is based on the Exponential Integrator designed for discretizing ordinary differential equations (ODEs) and leverages a semilinear structure of the learned diffusion process to reduce the discretization error. The proposed method can be applied to any DMs and can generate high-fidelity samples in as few as 10 steps. Moreover, by directly using pre-trained DMs, we achieve state-of-art sampling performance when the number of score function evaluation (NFE) is limited, e.g., 4.17 FID with 10 NFEs, 2.86 FID with only 20 NFEs on CIFAR10. Project page and code: https://qsh-zh.github.io/deis.

## 1 INTRODUCTION

The Diffusion model (DM) (Ho et al., 2020) is a generative modeling method developed recently that relies on the basic idea of reversing a given simple diffusion process. A time-dependent score function is learned for this purpose and DMs are thus also known as score-based models (Song et al., 2020b). Compared with other generative models such as generative adversarial networks (GANs), in addition to great scalability, the DM has the advantage of stable training is less hyperparameter sensitive (Creswell et al., 2018; Kingma & Welling, 2019). DMs have recently achieved impressive performances on a variety of tasks, including unconditional image generation (Ho et al., 2020; Song et al., 2020b; Rombach et al., 2021; Dhariwal & Nichol, 2021), text conditioned image generation (Nichol et al., 2021; Ramesh et al., 2022), text generation (Hoogeboom et al., 2021; Austin et al., 2021), 3D point cloud generation (Lyu et al., 2021), inverse problem (Kawar et al., 2021; Song et al., 2021b), etc.

However, the remarkable performance of DMs comes at the cost of slow sampling; it takes much longer time to produce high-quality samples compared with GANs. For instance, the Denoising Diffusion Probabilistic Model (DDPM) (Ho et al., 2020) needs 1000 steps to generate one sample and each step requires evaluating the learning neural network once; this is substantially slower than GANs (Goodfellow et al., 2014; Karras et al., 2019). For this reason, there exist several studies aiming at improve the sampling speed for DMs (More related works are discussed in App. A). One category of methods modify/optimize the forward noising process such that backward denoising process can be more efficient (Nichol & Dhariwal, 2021; Song et al., 2020b; Watson et al., 2021; Bao et al., 2022). An important and effective instance is the Denoising Diffusion Implicit Model (DDIM) (Song et al., 2020a) that uses a non-Markovian noising process. Another category of methods speed up the numerical solver for stochastic differential equations (SDEs) or ordinary differential equations (ODEs) associated with the DMs (Jolicoeur-Martineau et al., 2021; Song et al., 2020b; Tachibana et al., 2021). In (Song et al., 2020b), blackbox ODE solvers are used to solve a marginal equivalent ODE known as the Probability Flow (PF), for fast sampling. In (Liu et al.,

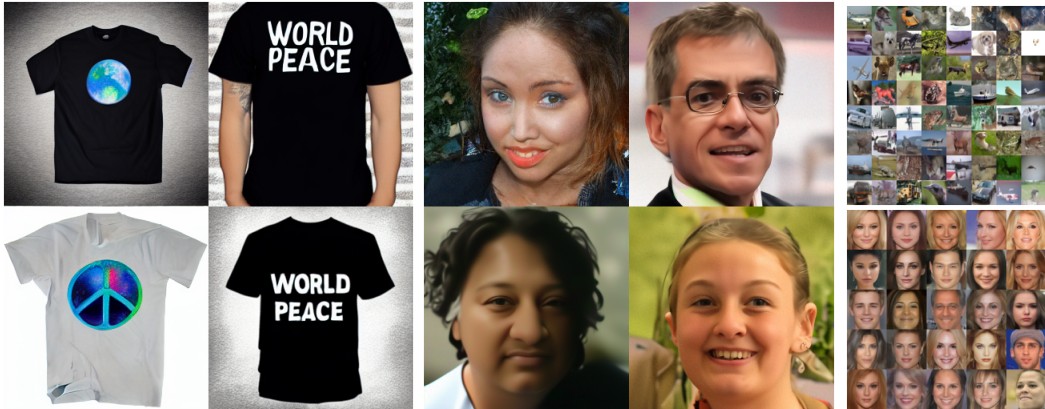

Figure 1: Generated images with various DMs. Latent diffusion (Rombach et al., 2021) (Left), $256 \times 256$ image with text *A shirt with inscription "World peace"* (15 NFE). VE diffusion (Song et al., 2020b) (Mid), FFHQ $256 \times 256$ (12 NFE). VP diffusion (Ho et al., 2020) (Right), CIFAR10 (7 NFE) and CELEBA (5 NFE).

2022), the authors combine DDIM with high order methods to solve this ODE and achieve further acceleration. Note that the deterministic DDIM can also be viewed as a time discretization of the PF as it matches the latter in the continuous limit (Song et al., 2020a; Liu et al., 2022). However, it is unclear why DDIM works better than generic methods such as Euler.

The objective of this work is to establish a principled discretization scheme for the learned backward diffusion processes in DMs so as to achieve fast sampling. Since the most expensive part in sampling a DM is the evaluation of the neural network that parameterizes the backward diffusion, we seek a discretization method that requires a small number of network function evaluation (NFE). We start with a family of marginal equivalent SDEs/ODEs associated with DMs and investigate numerical error sources, which include fitting error and discretization error. We observe that even with the same trained model, different discretization schemes can have dramatically different performances in terms of discretization error. We then carry out a sequence of experiments to systematically investigate the influences of different factors on the discretization error. We find out that the *Exponential Integrator (EI)* (Hochbruck & Ostermann, 2010) that utilizes the semilinear structure of the backward diffusion has minimum error. To further reduce the discretization error, we propose to either use high order polynomials to approximate the nonlinear term in the ODE or employ Runge Kutta methods on a transformed ODE. The resulting algorithms, termed *Diffusion Exponential Integrator Sampler (DEIS)*, achieve the best sampling quality with limited NFEs.

Our contributions are summarized as follows: 1) We investigate a family of marginal equivalent SDEs/ODEs for fast sampling and conduct a systematic error analysis for their numerical solvers. 2) We propose DEIS, an efficient sampler that can be applied to any DMs to achieve superior sampling quality with a limited number of NFEs. DEIS can also accelerate data log-likelihood evaluation. 3) We prove that the deterministic DDIM is a special case of DEIS, justifying the effectiveness of DDIM from a discretization perspective. 4) We conduct comprehensive experiments to validate the efficacy of DEIS. For instance, with a pre-trained model (Song et al., 2020b), DEIS is able to reach 4.17 FID with 10 NFEs, and 2.86 FID with 20 NFEs on CIFAR10.

## 2 BACKGROUND ON DIFFUSION MODELS

A DM consists of a fixed forward diffusion (noising) process that adds noise to the data, and a learned backward diffusion (denoising) process that gradually removes the added noise. The backward diffusion is trained to match the forward one in probability law, and when this happens, one can in principle generate perfect samples from the data distribution by simulating the backward diffusion.

**Forward noising diffusion**: The forward diffusion of a DM for $D$-dimensional data is a linear diffusion described by the stochastic differential equation (SDE) (Särkkä & Solin, 2019)

$$d\boldsymbol{x} = \boldsymbol{F}_t \boldsymbol{x} dt + \boldsymbol{G}_t d\boldsymbol{w}, \tag{1}$$

Table 1: Two popular SDEs, variance preserving SDE (VPSDE) and variance exploding SDE (VESDE). The parameter $\alpha_t$ is decreasing with $\alpha_0 \approx 1, \alpha_T \approx 0$, while $\sigma_t$ is increasing.

| SDE | $\boldsymbol{F}_t$ | $\boldsymbol{G}_t$ | $\mu_t$ | $\Sigma_t$ |
|---|---|---|---|---|
| VPSDE (Ho et al., 2020) | $\frac{1}{2}\frac{d\log\alpha_t}{dt}\boldsymbol{I}$ | $\sqrt{-\frac{d\log\alpha_t}{dt}}\boldsymbol{I}$ | $\sqrt{\alpha_t}\boldsymbol{I}$ | $(1-\alpha_t)\boldsymbol{I}$ |
| VESDE (Song et al., 2020b) | $0$ | $\sqrt{\frac{d[\sigma_t^2]}{dt}}\boldsymbol{I}$ | $\boldsymbol{I}$ | $\sigma_t^2\boldsymbol{I}$ |

where $\boldsymbol{F}_t \in \mathbb{R}^{D\times D}$ denotes the linear drift coefficient, $\boldsymbol{G}_t \in \mathbb{R}^{D\times D}$ denotes the diffusion coefficient, and $\boldsymbol{w}$ is a standard Wiener process. The diffusion Eq. (1) is initiated at the training data and simulated over a fixed time window $[0, T]$. Denote by $p_t(\boldsymbol{x}_t)$ the marginal distribution of $\boldsymbol{x}_t$ and by $p_{0t}(\boldsymbol{x}_t|\boldsymbol{x}_0)$ the conditional distribution from $\boldsymbol{x}_0$ to $\boldsymbol{x}_t$, then $p_0(\boldsymbol{x}_0)$ represents the underlying distribution of the training data. The simulated trajectories are represented by $\{\boldsymbol{x}_t\}_{0\le t\le T}$. The parameters $\boldsymbol{F}_t$ and $\boldsymbol{G}_t$ are chosen such that the conditional marginal distribution $p_{0t}(\boldsymbol{x}_t|\boldsymbol{x}_0)$ is a simple Gaussian distribution, denoted as $\mathcal{N}(\mu_t\boldsymbol{x}_0, \Sigma_t)$, and the distribution $\pi(\boldsymbol{x}_T) := p_T(\boldsymbol{x}_T)$ is easy to sample from. Two popular SDEs in diffusion models (Song et al., 2020b) are summarized in Tab. 1. Here we use matrix notation for $\boldsymbol{F}_t$ and $\boldsymbol{G}_t$ to highlight the generality of our method. Our approach is applicable to any DMs, including the Blurring diffusion models (BDM) (Hoogeboom & Salimans, 2022; Rissanen et al., 2022) and the critically-damped Langevin diffusion (CLD) (Dockhorn et al., 2021) where these coefficients are indeed non-diagonal matrices.

**Backward denoising diffusion**: Under mild assumptions (Anderson, 1982; Song et al., 2020b), the forward diffusion Eq. (1) is associated with a reverse-time diffusion process

$$d\boldsymbol{x} = [\boldsymbol{F}_t\boldsymbol{x}dt - \boldsymbol{G}_t\boldsymbol{G}_t^T\nabla\log p_t(\boldsymbol{x})]dt + \boldsymbol{G}_t d\boldsymbol{w}, \tag{2}$$

where $\boldsymbol{w}$ denotes a standard Wiener process in the reverse-time direction. The distribution of the trajectories of Eq. (2) with terminal distribution $\boldsymbol{x}_T \sim \pi$ coincides with that of Eq. (1) with initial distribution $\boldsymbol{x}_0 \sim p_0$, that is, Eq. (2) matches Eq. (1) in probability law. Thus, in principle, we can generate new samples from the data distribution $p_0$ by simulating the backward diffusion Eq. (2). However, to solve Eq. (2), we need to evaluate the score function $\nabla\log p_t(\boldsymbol{x})$, which is not accessible.

**Training**: The basic idea of DMs is to use a time-dependent network $\boldsymbol{s}_\theta(\boldsymbol{x}, t)$, known as a score network, to approximate the score $\nabla\log p_t(\boldsymbol{x})$. This is achieved by score matching techniques (Hyvärinen, 2005; Vincent, 2011) where the score network $\boldsymbol{s}_\theta$ is trained by minimizing the denoising score matching loss

$$\mathcal{L}(\theta) = \mathbb{E}_{t\sim\text{Unif}[0,T]}\mathbb{E}_{p(\boldsymbol{x}_0)p_{0t}(\boldsymbol{x}_t|\boldsymbol{x}_0)}[\|\nabla\log p_{0t}(\boldsymbol{x}_t|\boldsymbol{x}_0) - \boldsymbol{s}_\theta(\boldsymbol{x}_t, t)\|_{\Lambda_t}^2]. \tag{3}$$

Here $\nabla\log p_{0t}(\boldsymbol{x}_t|\boldsymbol{x}_0)$ has a closed form expression as $p_{0t}(\boldsymbol{x}_t|\boldsymbol{x}_0)$ is a simple Gaussian distribution, and $\Lambda_t$ denotes a time-dependent weight. This loss can be evaluated using empirical samples by Monte Carlo methods and thus standard stochastic optimization algorithms can be used for training. We refer the reader to (Ho et al., 2020; Song et al., 2020b) for more details on choices of $\Lambda_t$ and training techniques.

## 3 FAST SAMPLING WITH LEARNED SCORE MODELS

Once the score network $\boldsymbol{s}_\theta(\boldsymbol{x}, t) \approx \nabla\log p_t(\boldsymbol{x})$ is trained, it can be used to generate new samples by solving the backward SDE Eq. (2) with $\nabla\log p_t(\boldsymbol{x})$ replaced by $\boldsymbol{s}_\theta(\boldsymbol{x}, t)$. It turns out there are infinitely many diffusion processes one can use. In this work, we consider a family of SDEs

$$d\hat{\boldsymbol{x}} = [\boldsymbol{F}_t\hat{\boldsymbol{x}} - \frac{1+\lambda^2}{2}\boldsymbol{G}_t\boldsymbol{G}_t^T\boldsymbol{s}_\theta(\hat{\boldsymbol{x}}, t)]dt + \lambda\boldsymbol{G}_t d\boldsymbol{w}, \tag{4}$$

parameterized by $\lambda \ge 0$. Here we use $\hat{\boldsymbol{x}}$ to distinguish the solution to the SDE associated with the learned score from the ground truth $\boldsymbol{x}$ in Eqs. (1) and (2). When $\lambda = 0$, Eq. (4) reduces to an ODE known as the *probability flow ODE* (Song et al., 2020b). The reverse-time diffusion Eq. (2) with an approximated score is a special case of Eq. (4) with $\lambda = 1$. Denote the trajectories generated by Eq. (4) as $\{\hat{\boldsymbol{x}}_t^*\}_{0\le t\le T}$ and the marginal distributions as $\hat{p}_t^*$. The following Proposition (Zhang & Chen, 2021) (Proof in App. D) holds.

**Proposition 1.** *When $s_\theta(x, t) = \nabla \log p_t(x)$ for all $x, t$, and $\hat{p}_T^* = \pi$, the marginal distribution $\hat{p}_t^*$ of Eq. (4) matches $p_t$ of the forward diffusion Eq. (1) for all $0 \le t \le T$.*

The above result justifies the usage of Eq. (4) for generating samples. To generate a new sample, one can sample $\hat{x}_T^*$ from $\pi$ and solve Eq. (4) to obtain a sample $\hat{x}_0^*$. However, in practice, exact solutions to Eq. (4) are not attainable and one needs to discretize Eq. (4) over time to get an approximated solution. Denote the approximated solution by $\hat{x}_t$ and its marginal distribution by $\hat{p}_t$, then the error of the generative model, that is, the difference between $p_0(x)$ and $\hat{p}_0(x)$, is caused by two error sources, *fitting error* and *discretization error*. The fitting error is due to the mismatch between the learned score network $s_\theta$ and the ground truth score $\nabla \log p_t(x)$. The discretization error includes all extra errors introduced by the discretization in numerically solving Eq. (4). To reduce discretization error, one needs to use smaller stepsize and thus larger number of steps, making the sampling less efficient.

The objective of this work is to investigate these two error sources and develop a more efficient sampling scheme from Eq. (4) with less errors. In this section, we focus on the ODE approach with $\lambda = 0$. All experiments in this section are conducted based on VPSDE over the CIFAR10 dataset unless stated otherwise. The discussions on SDE approach with $\lambda > 0$ are deferred to App. C.

### 3.1 CAN WE LEARN GLOBALLY ACCURATE SCORE?

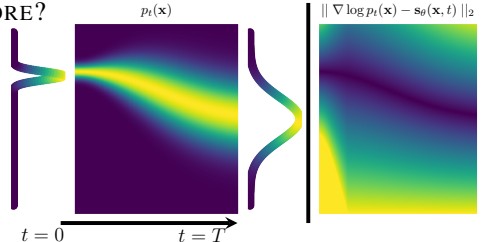

Since DMs demonstrate impressive empirical results in generating high-fidelity samples, it is tempting to believe that the learned score network is able to fit the score of data distribution very well, that is, $s_\theta(x, t) \approx \nabla \log p_t(x)$ for almost all $x \in \mathbb{R}^D$ and $t \in [0, T]$. This is, however, not true; the fitting error can be arbitrarily large on some $x, t$ as illustrated in a simple example below. In fact, the learned score models are not accurate for most $x, t$.

Figure 2: Fitting error on a toy demo. Lighter areas represent higher probability region (left) and larger fitting error (right).

Consider a generative modeling task over 1-dimensional space, i.e., $D = 1$. The data distribution is a Gaussian concentrated with a very small variance. We plot the fitting error[1] between a score model trained by minimizing Eq. (3) and the ground truth score in Fig. 2. As can be seen from the figure, the score model works well in the region where $p_t(x)$ is large but suffers from large error in the region where $p_t(x)$ is small. This observation can be explained by examining the training loss Eq. (3). In particular, the training data of Eq. (3) are sampled from $p_t(x)$. In regions with a low $p_t(x)$ value, the learned score network is not expected to work well due to the lack of training data. This phenomenon becomes even clearer in realistic settings with high-dimensional data. The region with high $p_t(x)$ value is extremely small since realistic data is often sparsely distributed in $\mathbb{R}^D$; it is believed real data such as images concentrate on an intrinsic low dimensional manifold (Deng et al., 2009; Pless & Souvenir, 2009; Liu et al., 2022).

As a consequence, to ensure $\hat{x}_0$ is close to $x_0$, we need to make sure $\hat{x}_t$ stays in the high $p_t(x)$ region for all $t$. This makes fast sampling from Eq. (4) a challenging task as it prevents us from taking an aggressive step size that is likely to take the solution to the region where the fitting error of the learned score network is large. A good discretization scheme for Eq. (4) should be able to help reduce the impact of the fitting error of the score network during sampling.

### 3.2 DISCRETIZATION ERROR

We next investigate the discretization error of solving the probability flow ODE ($\lambda = 0$)

$$\frac{d\hat{x}}{dt} = F_t \hat{x} - \frac{1}{2} G_t G_t^T s_\theta(\hat{x}, t). \tag{5}$$

The exact solution to this ODE is

$$\hat{x}_t = \Psi(t, s)\hat{x}_s + \int_s^t \Psi(t, \tau)[-\frac{1}{2} G_\tau G_\tau^T s_\theta(\hat{x}_\tau, \tau)]d\tau, \tag{6}$$

---

[1]Because the fitting error explodes when $t \to 0$, we have scaled the fitting error for better visualization.

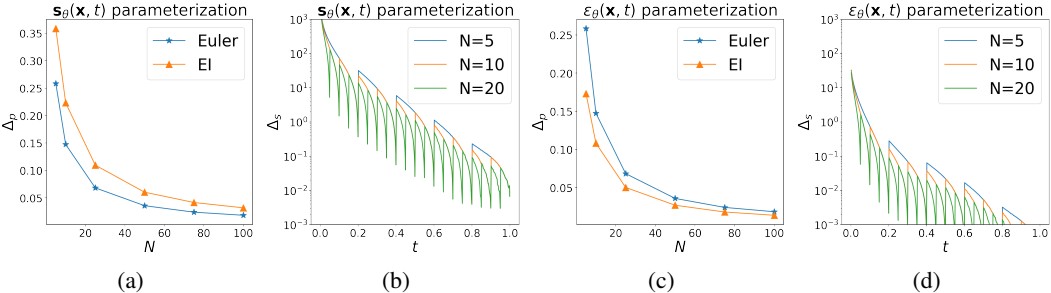

Figure 3: Fig. 3a shows average pixel difference $\Delta_p$ between ground truth $\hat{x}_0^*$ and numerical solution $\hat{x}_0$ from Euler method and EI method. Fig. 3b depicts approximation error $\Delta_s$ along ground truth solutions. Fig. 3d shows $\Delta_s$ can be dramatically reduced if the parameterization $\epsilon_\theta(x, t)$ instead of $s_\theta(x, t)$ is used. This parameterization helps the EI method outperform the Euler method in Fig. 3c.

where $\Psi(t, s)$ satisfying $\frac{\partial}{\partial t}\Psi(t, s) = F_t\Psi(t, s), \Psi(s, s) = I$ is known as the transition matrix from time $s$ to $t$ associated with $F_\tau$. Eq. (5) is a semilinear stiff ODE (Hochbruck & Ostermann, 2010) that consists of a linear term $F_t\hat{x}$ and a nonlinear term $s_\theta(\hat{x}, t)$. There exist many different numerical solvers for Eq. (5) associated with different discretization schemes to approximate Eq. (6) (Griffiths & Higham, 2010). As the discretization step size goes to zero, the solutions obtained from all these methods converge to that of Eq. (5). However, the performances of these methods can be dramatically different when the step size is large. On the other hand, to achieve fast sampling with Eq. (5), we need to approximately solve it with a small number of discretization steps, and thus large step size. This motivates us to develop an efficient discretizaiton scheme that fits with Eq. (5) best. In the rest of this section, we systematically study the discretization error in solving Eq. (5), both theoretically and empirically with carefully designed experiments. Based on these results, we develop an efficient algorithm for Eq. (5) that requires a small number of NFEs.

**Ingredient 1: Exponential Integrator over Euler method.** The Euler method is the most elementary explicit numerical method for ODEs and is widely used in numerical softwares (Virtanen et al., 2020). When applied to Eq. (5), the Euler method reads

$$\hat{x}_{t-\Delta t} = \hat{x}_t - [F_t\hat{x}_t - \frac{1}{2}G_tG_t^T s_\theta(\hat{x}_t, t)]\Delta t. \tag{7}$$

This is used in many existing works in DMs (Song et al., 2020b; Dockhorn et al., 2021). This approach however has low accuracy and is sometimes unstable when the stepsize is not sufficiently small. To improve the accuracy, we propose to use the *Exponential Integrator (EI)*, a method that leverages the semilinear structure of Eq. (5). When applied to Eq. (5), the EI reads

$$\hat{x}_{t-\Delta t} = \Psi(t - \Delta t, t)\hat{x}_t + [\int_t^{t-\Delta t} -\frac{1}{2}\Psi(t - \Delta t, \tau)G_\tau G_\tau^T d\tau]s_\theta(\hat{x}_t, t). \tag{8}$$

It is effective if the nonlinear term $s_\theta(\hat{x}_t, t)$ does not change much along the solution. In fact, for any given $\Delta t$, Eq. (8) solves Eq. (5) exactly if $s_\theta(\hat{x}_t, t)$ is constant over the time interval $[t - \Delta t, t]$.

To compare the EI Eq. (8) and the Euler method Eq. (7), we plot in Fig. 3a the *average pixel difference* $\Delta_p$ between the ground truth $\hat{x}_0^*$ and the numerical solution $\hat{x}_0$ obtained by these two methods for various number $N$ of steps. Surprisingly, the EI method performs worse than the Euler method.

This observation suggests that there are other major factors that contribute to the error $\Delta_p$. In particular, the condition that the nonlinear term $s_\theta(\hat{x}_t, t)$ does not change much along the solution assumed for the EI method does not hold. To see this, we plot the *score approximation error* $\Delta_s(\tau) = ||s_\theta(x_\tau, \tau) - s_\theta(x_t, t)||_2, \tau \in [t - \Delta t, t]$ along the exact solution $\{\hat{x}_t^*\}$ to Eq. (5) in Fig. 3b[2]. It can be seen that the approximation error grows rapidly as $t$ approaches 0. This is not strange; the score of realistic data distribution $\nabla \log p_t(x)$ should change rapidly as $t \to 0$ (Dockhorn et al., 2021).

---

[2]The $\{\hat{x}_t^*\}$ are approximated by solving ODE with high accuracy solvers and sufficiently small step size. For better visualization, we have removed the time discretization points in Fig. 3b and Fig. 3d, since $\Delta_s = 0$ at these points and becomes negative infinity in log scale.

**Ingredient 2: $\epsilon_\theta(\boldsymbol{x}, t)$ over $\boldsymbol{s}_\theta(\boldsymbol{x}, t)$.** The issues caused by rapidly changing score $\nabla \log p_t(\boldsymbol{x})$ do not only exist in sampling, but also appear in the training of DMs. To address these issues, a different parameterization of the score network is used. In particular, it is found that the parameterization (Ho et al., 2020) $\nabla \log p_t(\boldsymbol{x}) \approx -\boldsymbol{L}_t^{-T}\epsilon_\theta(\boldsymbol{x}, t)$, where $\boldsymbol{L}_t$ can be any matrix satisfying $\boldsymbol{L}_t\boldsymbol{L}_t^T = \Sigma_t$, leads to significant improvements of accuracy. The rationale of this parameterization is based on a reformulation of the training loss Eq. (3) as (Ho et al., 2020)

$$\bar{\mathcal{L}}(\theta) = \mathbb{E}_{t \sim \text{Unif}[0,T]}\mathbb{E}_{p(\boldsymbol{x}_0), \epsilon \sim \mathcal{N}(0, \boldsymbol{I})}[\|\epsilon - \epsilon_\theta(\mu_t\boldsymbol{x}_0 + \boldsymbol{L}_t\epsilon, t)\|_{\bar{\Lambda}_t}^2] \tag{9}$$

with $\bar{\Lambda}_t = \boldsymbol{L}_t^{-1}\Lambda_t\boldsymbol{L}_t^{-T}$. The network $\epsilon_\theta$ tries to follow $\epsilon$ which is sampled from a standard Gaussian and thus has a small magnitude. In comparison, the parameterization $\boldsymbol{s}_\theta = -\boldsymbol{L}_t^{-T}\epsilon_\theta$ can take large value as $\boldsymbol{L}_t \to 0$ as $t$ approaches 0. It is thus better to approximate $\epsilon_\theta$ than $\boldsymbol{s}_\theta$ with a neural network.

We adopt this parameterization and rewrite Eq. (5) as

$$\frac{d\hat{\boldsymbol{x}}}{dt} = \boldsymbol{F}_t\hat{\boldsymbol{x}} + \frac{1}{2}\boldsymbol{G}_t\boldsymbol{G}_t^T\boldsymbol{L}_t^{-T}\epsilon_\theta(\hat{\boldsymbol{x}}, t). \tag{10}$$

Applying the EI to Eq. (10) yields

$$\hat{\boldsymbol{x}}_{t-\Delta t} = \Psi(t - \Delta t, t)\hat{\boldsymbol{x}}_t + [\int_t^{t-\Delta t} \frac{1}{2}\Psi(t - \Delta t, \tau)\boldsymbol{G}_\tau\boldsymbol{G}_\tau^T\boldsymbol{L}_\tau^{-T}d\tau]\epsilon_\theta(\hat{\boldsymbol{x}}_t, t). \tag{11}$$

Compared with Eq. (8), Eq. (11) employs $-\boldsymbol{L}_\tau^{-T}\epsilon_\theta(\boldsymbol{x}_t, t)$ instead of $\boldsymbol{s}_\theta(\boldsymbol{x}_t, t) = -\boldsymbol{L}_t^{-T}\epsilon_\theta(\boldsymbol{x}_t, t)$ to approximate the score $\boldsymbol{s}_\theta(\boldsymbol{x}_\tau, \tau)$ over the time interval $\tau \in [t - \Delta t, t]$. This modification from $\boldsymbol{L}_t^{-T}$ to $\boldsymbol{L}_\tau^{-T}$ turns out to be crucial; the coefficient $\boldsymbol{L}_\tau^{-T}$ changes rapidly over time. This is verified by Fig. 3d where we plot the score approximation error $\Delta_s$ when the parameterization $\epsilon_\theta$ is used, from which we see the error $\Delta_s$ is greatly reduced compared with Fig. 3b. With this modification, the EI method significantly outperforms the Euler method as shown in Fig. 3c. Next we develop several fast sampling algorithms, all coined as the *Diffusion Exponential Integrator Sampler (DEIS)*, based on Eq. (11), for DMs.

Interestingly, the discretization Eq. (11) based on EI coincides with the popular deterministic DDIM when the forward diffusion Eq. (1) is VPSDE (Song et al., 2020a) as summarized below (Proof in App. E).

**Proposition 2.** *When the forward diffusion Eq. (1) is set to be VPSDE ($\boldsymbol{F}_t, \boldsymbol{G}_t$ are specified in Tab. 1), the EI discretization Eq. (11) becomes*

$$\hat{\boldsymbol{x}}_{t-\Delta t} = \sqrt{\frac{\alpha_{t-\Delta t}}{\alpha_t}}\hat{\boldsymbol{x}}_t + [\sqrt{1 - \alpha_{t-\Delta t}} - \sqrt{\frac{\alpha_{t-\Delta t}}{\alpha_t}}\sqrt{1 - \alpha_t}]\epsilon_\theta(\hat{\boldsymbol{x}}_t, t), \tag{12}$$

*which coincides with the deterministic DDIM sampling algorithm.*

Our result provides an alternative justification for the efficacy of DDIM for VPSDE from a numerical discretization point of view. Unlike DDIM, our method Eq. (11) can be applied to any diffusion SDEs to improve the efficiency and accuracy of discretizations.

In the discretization Eq. (11), we use $\epsilon_\theta(\hat{\boldsymbol{x}}_t, t)$ to approximate $\epsilon_\theta(\hat{\boldsymbol{x}}_\tau, \tau)$ for all $\tau \in [t - \Delta t, t]$, which is a zero order approximation. Comparing Eq. (11) and Eq. (6) we see that this approximation error largely determines the accuracy of discretization. One natural question to ask is whether it is possible to use a better approximation of $\epsilon_\theta(\hat{\boldsymbol{x}}_\tau, \tau)$ to further improve the accuracy? We answer this question affirmatively below with an improved algorithm.

**Ingredient 3: Polynomial extrapolation of $\epsilon_\theta$.** Before presenting our algorithm, we investigate how $\epsilon_\theta(\boldsymbol{x}_t, t)$ evolves along a ground truth solution $\{\hat{\boldsymbol{x}}_t\}$ from $t = T$ to $t = 0$. We plot the relative change in 2-norm of $\epsilon_\theta(\boldsymbol{x}_t, t)$ in Fig. 4a. It reveals that for most time instances the relative change is small. This motivates us to use previous (backward) evaluations of $\epsilon_\theta$ up to $t$ to extrapolate $\epsilon_\theta(\boldsymbol{x}_\tau, \tau)$ for $\tau \in [t - \Delta t, t]$.

Inspired by the high-order polynomial extrapolation in linear multistep methods, we propose to use high-order polynomial extrapolation of $\epsilon_\theta$ in our EI method. To this end, consider time discretization $\{t_i\}_{i=0}^N$ where $t_0 = 0, t_N = T$. For each $i$, we fit a polynomial $\boldsymbol{P}_r(t)$ of degree $r$ with respect to the interpolation points $(t_{i+j}, \epsilon_\theta(\hat{\boldsymbol{x}}_{t_{i+j}}, t_{i+j})), 0 \leq j \leq r$. This polynomial $\boldsymbol{P}_r(t)$ has explicit expression

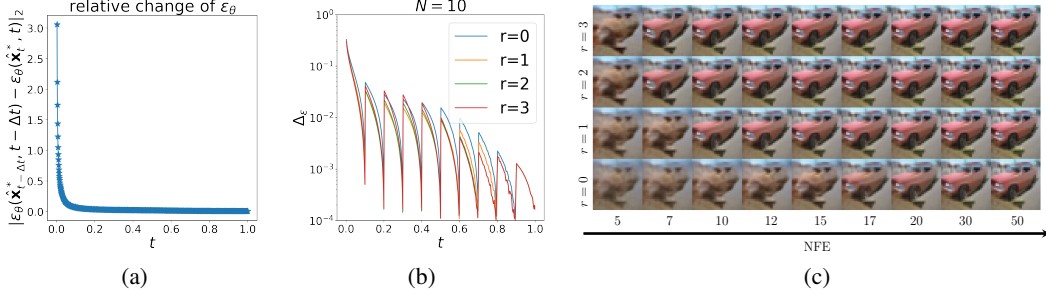

(a)            (b)            (c)

Figure 4: Fig. 4a shows relative changes of $\epsilon_\theta(\hat{\boldsymbol{x}}_t^*, t)$ with respect to $t$ are relatively small, especially when $t > 0.15$. Fig. 4b depicts the extrapolation error with $N = 10$. High order polynomial can reduce approximation error effectively. Fig. 4c illustrates effects of extrapolation. When $N$ is small, higher order polynomial approximation leads to better samples.

$$\boldsymbol{P}_r(t) = \sum_{j=0}^{r}[\prod_{k \neq j}\frac{t - t_{i+k}}{t_{i+j} - t_{i+k}}]\epsilon_\theta(\hat{\boldsymbol{x}}_{t_{i+j}}, t_{i+j}). \tag{13}$$

We then use $\boldsymbol{P}_r(t)$ to approximate $\epsilon_\theta(\boldsymbol{x}_\tau, \tau)$ over the interval $[t_{i-1}, t_i]$. For $i > N - r$, we need to use polynomials of lower order to approximate $\epsilon_\theta$. To see the advantages of this approximation, we plot the approximate error $\Delta_\epsilon(t) = ||\epsilon_\theta(\boldsymbol{x}_t, t) - \boldsymbol{P}_r(t)||_2$ of $\epsilon_\theta(\boldsymbol{x}_t, t)$ by $\boldsymbol{P}_r(t)$ along ground truth trajectories $\{\hat{\boldsymbol{x}}_t^*\}$ in Fig. 4b. It can be seen that higher order polynomials can reduce approximation error compared with the case $r = 0$ which uses zero order approximation as in Eq. (11).

As in the EI method Eq. (11) that uses a zero order approximation of the score in Eq. (6), the update step of order $r$ is obtained by plugging the polynomial approximation Eq. (13) into Eq. (6). It can be written explicitly as

$$\hat{\boldsymbol{x}}_{t_{i-1}} = \Psi(t_{i-1}, t_i)\hat{\boldsymbol{x}}_{t_i} + \sum_{j=0}^{r}[C_{ij}\epsilon_\theta(\hat{\boldsymbol{x}}_{t_{i+j}}, t_{i+j})] \tag{14}$$

$$C_{ij} = \int_{t_i}^{t_{i-1}} \frac{1}{2}\Psi(t_{i-1}, \tau)\boldsymbol{G}_\tau\boldsymbol{G}_\tau^T\boldsymbol{L}_\tau^{-T}\prod_{k \neq j}[\frac{\tau - t_{i+k}}{t_{i+j} - t_{i+k}}]d\tau. \tag{15}$$

We remark that the update in Eq. (14) is a linear combination of $\hat{\boldsymbol{x}}_{t_i}$ and $\epsilon_\theta(\hat{\boldsymbol{x}}_{t_{i+j}}, t_{i+j})$, where the weights $\Psi(t_{i-1}, t_i)$ and $C_{ij}$ are calculated once for a given forward diffusion Eq. (1) and time discretization, and can be reused across batches. For some diffusion Eq. (1), $\Psi(t_{i-1}, t_i), C_{ij}$ have closed form expression. Even if analytic formulas are not available, one can use high accuracy solver to obtain these coefficients. In DMs (e.g., VPSDE and VESDE), Eq. (15) are normally 1-dimensional or 2-dimensional integrations and are thus easy to evaluate numerically. This approach resembles the classical Adams–Bashforth (Hochbruck & Ostermann, 2010) method, thus we term it $t$AB-DEIS. Here we use $t$ to differentiate it from other DEIS algorithms we present later in Sec. 4 based on a time-scaled ODE.

The $t$AB-DEIS algorithm is summarized in Algo 1. Note that the deterministic DDIM is a special case of $t$AB-DEIS for VPSDE with $r = 0$. The polynomial approximation used in DEIS improves the sampling quality significantly when sampling steps $N$ is small, as shown in Fig. 4c.

## 4    EXPONENTIAL INTEGRATOR: SIMPLIFY PROBABILITY FLOW ODE

Next we present a different perspective to DEIS based on ODE transformations. The probability ODE Eq. (10) can be transformed into a simple non-stiff ODE, and then off-the-shelf ODE solvers can be applied to solve the ODE effectively. To this end, we introduce variable $\hat{\boldsymbol{y}}_t := \Psi(t, 0)\hat{\boldsymbol{x}}_t$ and rewrite Eq. (10) into

$$\frac{d\hat{\boldsymbol{y}}}{dt} = \frac{1}{2}\Psi(t, 0)\boldsymbol{G}_t\boldsymbol{G}_t^T\boldsymbol{L}_t^{-T}\epsilon_\theta(\Psi(0, t)\hat{\boldsymbol{y}}, t). \tag{16}$$

Note that, departing from Eq. (10), Eq. (16) does not possess semi-linear structure. Thus, we can apply off-the-shelf ODE solvers to Eq. (16) without accounting for the semi-linear structure in algorithm design. This transformation Eq. (16) can be further improved by taking into account the

**Algorithm 1** $t$AB-DEIS

**Input**: $\{t_i\}_{i=0}^N, r$
**Instantiate**: $\hat{\boldsymbol{x}}_{t_N}$, Empty $\epsilon$-buffer
Calculate weights $\Psi, \boldsymbol{C}$ based on Eq. (15)
**for** $i$ in $N, N-1, \cdots, 1$ **do**
    $\epsilon$-buffer.append($\epsilon_\theta(\hat{\boldsymbol{x}}_{t_i}, t_i)$)
    $\hat{\boldsymbol{x}}_{t_{i-1}} \leftarrow$ Eq. (14) with $\Psi, \boldsymbol{C}, \epsilon$-buffer
**end for**

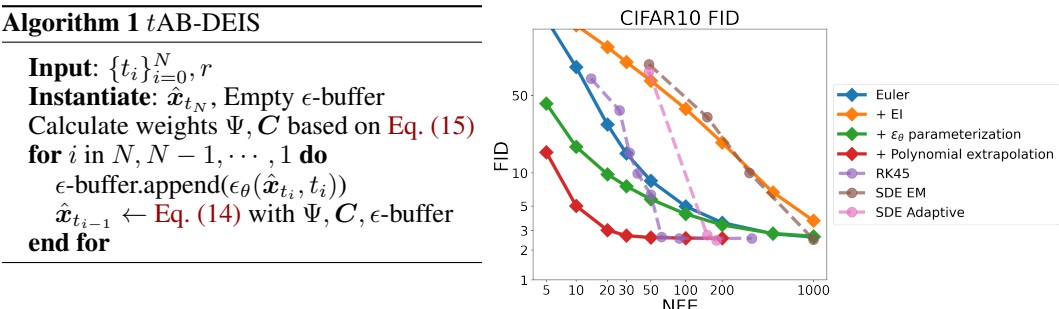

Figure 5: Ablation study and comparison with other samplers. We notice switching from Euler to Exponential Integrator worsens FID, which we explore and explain Ingredient 2 in Sec. 3. With EI, $\epsilon_\theta$, polynomial extrapolation and optimizing timestamps can significantly improve the sampling quality. Compared with other samplers, ODE sampler based on RK45 (Song et al., 2020b), SDE samplers based on Euler-Maruyama (EM) (Song et al., 2020b) and SDE adaptive step size solver (Jolicoeur-Martineau et al., 2021), DEIS can converge much faster.

analytical form of $\Psi, \boldsymbol{G}_t, \boldsymbol{L}_t$. Here we present treatment for VPSDE; the results can be extended to other (scalar) DMs such as VESDE.

**Proposition 3.** *For the VPSDE, with $\hat{\boldsymbol{y}}_t = \sqrt{\frac{\alpha_0}{\alpha_t}}\hat{\boldsymbol{x}}_t$ and the time-scaling $\beta(t) = \sqrt{\alpha_0}(\sqrt{\frac{1-\alpha_t}{\alpha_t}} - \sqrt{\frac{1-\alpha_0}{\alpha_0}})$, Eq. (10) can be transformed into*

$$\frac{d\hat{\boldsymbol{y}}}{d\rho} = \epsilon_\theta(\sqrt{\frac{\alpha_{\beta^{-1}(\rho)}}{\alpha_0}}\hat{\boldsymbol{y}}, \beta^{-1}(\rho)), \quad \rho \in [\beta(0), \beta(T)]. \tag{17}$$

After transformation, the ODE becomes a black-box ODE that can be solved by generic ODE solvers efficiently since the stiffness caused by the semi-linear structure is removed. This is the core idea of the variants of DEIS we present next.

Based on the transformed ODE Eq. (17) and the above discussions, we propose two variants of the DEIS algorithm: $\rho$**RK-DEIS** when applying classical RK methods, and $\rho$**AB-DEIS** when applying Adams-Bashforth methods. We remark that the difference between $t$AB-DEIS and $\rho$AB-DEIS lies in the fact that $t$AB-DEIS fits polynomials in $t$ which may not be polynomials in $\rho$. Thanks to simplified ODEs, DEIS enjoys the convergence order guarantee as its underlying RK or AB solvers.

## 5 EXPERIMENTS

**Abalation study:** As shown in Fig. 5, ingredients introduced in Sec. 3.2 can significantly improve sampling efficiency on CIFAR10. Besides, DEIS outperforms standard samplers by a large margin.

**DEIS variants:** We include performance evaluations of various DEIS with VPSDE on CIFAR10 in Tab. 2, including DDIM, $\rho$RK-DEIS, $\rho$AB-DEIS and $t$AB-DEIS. For $\rho$RK-DEIS, we find Heun's method works best among second-order RK methods, denoted as $\rho$2Heun, Kutta method for third order, denoted as $\rho$3Kutta, and classic fourth-order RK denoted as $\rho$4RK. For Adam-Bashforth methods, we consider fitting $1, 2, 3$ order polynomial in $t, \rho$, denoted as $t$AB and $\rho$AB respectively. We observe that almost all DEIS algorithms can generate high-fidelity images with small NFE. Also, note that DEIS with high-order polynomial approximation can significantly outperform DDIM; the latter coincides with the zero-order polynomial approximation. We also find the performance of high order $\rho$RK-DEIS is not satisfying when NFE is small but competitive as NFE increases. It is within expectation as high order methods enjoy smaller local truncation error and total accumulated error when small step size is used and the advantage is vanishing as we reduce the number of steps.

**More comparisons**: We conduct more comparisons with popular sampler for DMs, including DDPM, DDIM, PNDM (Liu et al., 2021), A-DDIM (Bao et al., 2022), FastDPM (Kong & Ping, 2021), and Ito-Taylor (Tachibana et al., 2021). We further propose Improved PNDM (iPNDM) that avoids the expensive warming start, which leads to better empirical performance. We conduct

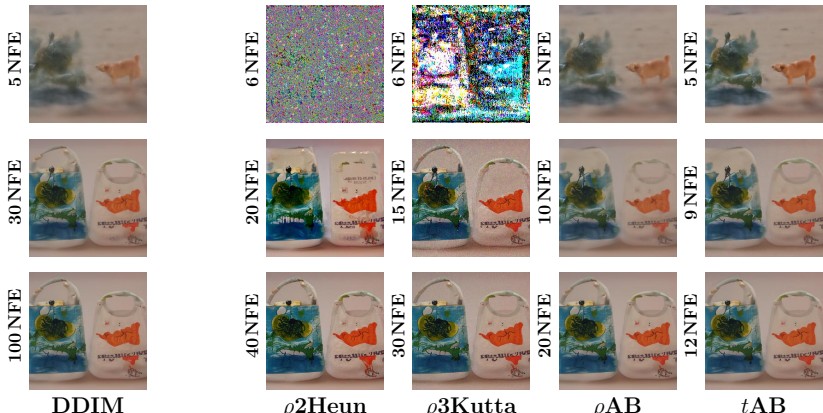

Figure 6: Generated samples of DDIM and DEIS with unconditional $256 \times 256$ ImageNet pretrained model (Dhariwal & Nichol, 2021)

| NFE | DDIM | $\rho$2Heun$^\dagger$ | $\rho$3Kutta | FID for various DEIS $\rho$4RK | $\rho$AB1 | $\rho$AB2 | $\rho$AB3 | $t$AB1 | $t$AB2 | $t$AB3 |
|---|---|---|---|---|---|---|---|---|---|---|
| 5  | 26.91 | $108^{+1}$   | $185^{+1}$    | $193^{+3}$    | 22.28 | 21.53 | 21.43 | 19.72 | 16.31 | **15.37** |
| 10 | 11.14 | 14.72        | $13.19^{+2}$  | $28.65^{+2}$  | 7.56  | 6.72  | 6.50  | 6.09  | 4.57  | **4.17**  |
| 15 | 7.06  | $4.89^{+1}$  | 5.88          | $6.88^{+1}$   | 4.69  | 4.16  | 3.99  | 4.29  | 3.57  | **3.37**  |
| 20 | 5.47  | 3.50         | $2.97^{+1}$   | 3.92          | 3.70  | 3.32  | 3.17  | 3.54  | 3.05  | **2.86**  |
| 50 | 3.27  | 2.60         | **$2.55^{+1}$** | $2.57^{+2}$ | 2.70  | 2.62  | 2.59  | 2.67  | 2.59  | 2.57      |

Table 2: More results of DEIS for VPSDE on CIFAR10 with limited NFE. For $\rho$RK-DEIS, the upper right number indicates extra NFEs used. Bold numbers denote the best performance achieved with similar NFE budgets. For a fair comparison, we report numbers based on their best time discretization for different algorithms with different NFE. We include a comparison given the same time discretization in App. H.3. $\dagger$: The concurrent work (Karras et al., 2022) applies Heun method to a rescaled DM. This is a special case of $\rho$2Heun (More discussion included in App. B).

comparison on image datasets, including $64 \times 64$ CelebA (Liu et al., 2015) with pre-trained model from Song et al. (2020a), class-conditioned $64 \times 64$ ImageNet (Deng et al., 2009) with pre-trained model (Dhariwal & Nichol, 2021), $256 \times 256$ LSUN Bedroom (Yu et al., 2015) with pre-trained model (Dhariwal & Nichol, 2021). We compare DEIS with selected baselines in Fig. 7 quantitatively, and show empirical samples in Fig. 6. More implementation details, the performance of various DMs, and many more qualitative experiments are included in App. H.

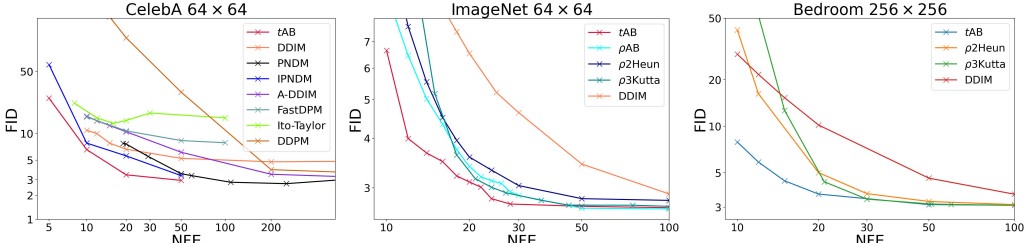

Figure 7: Sample quality measured by FID $\downarrow$ of different sampling algorithms with pre-trained DMs.

## 6 CONCLUSION

In this work, we consider fast sampling problems for DMs. We present the diffusion exponential integrator sampler (DEIS), a fast sampling algorithm for DMs based on a novel discretization scheme of the backward diffusion process. In addition to its theoretical elegance, DEIS also works efficiently in practice; it is able to generate high-fidelity samples with less than 10 NFEs. Exploring better extrapolation may further improve sampling quality. More discussions are included in App. B.

ACKNOWLEDGMENTS

The authors would like to thank the anonymous reviewers for their useful comments. This work is partially supported by NSF ECCS-1942523 and NSF CCF-2008513.

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

## A    MORE RELATED WORKS

A lot of research has been conducted to speed up the sampling of DMs. In (Kong & Ping, 2021; Watson et al., 2021) the authors optimize denosing process by modifying the underlying stochastic process. However, such acceleration can not generate high quality samples with a small number of discretization steps. In (Song et al., 2020a) the authors use a non-Markovian forward noising. The resulted algorihtm, DDIM, achieves significant acceleration than DDPMs. More recently, the authors of (Bao et al., 2022) optimize the backward Markovian process to approximate the non-Markovian forward process and get an analytic expression of optimal variance in denoising process. Another strategy to make the forward diffusion nonlinear and trainable (Zhang & Chen, 2021; Vargas et al., 2021; De Bortoli et al., 2021; Wang et al., 2021; Chen et al., 2021a) in the spirit of Schrödinger bridge (Chen et al., 2021b). This however comes with a heavy training overhead.

More closely related to our method is (Liu et al., 2022), which interprets update step in deterministic DDIM as a combination of gradient estimation step and transfer step. It modifies high order ODE methods to provide an estimation of the gradient and uses DDIM for transfer step. However, the decomposition of DDIM into two separate components is not theoretically justified. Based on our analysis on Exponential Integrator, Liu et al. (2022) uses Exponential Integral but with a Euler discretization-based approximation of the nonlinear term. This approximation is inaccurate and may suffer large discretization error if the step size is large as we show in Sec. 5.

The semilinear structure presented in probability flow ODE has been widely investigated in physics and numerical simulation (Hochbruck & Ostermann, 2010; Whalen et al., 2015), from which we get inspirations. The stiff property of the ODEs requires more efficient ODE solvers instead of black-box solvers that are designed for general ODE problems. In this work, we investigate sovlers for differential equations in diffusion model and take advantage of the semilinear structure.

## B    DISCUSSIONS

1. Q — *Can DEIS help accelerate the likelihood evaluation of diffusion models?*

   A — Theoretically, our methods can be used in likelihood evaluation as DEIS only changes numerical discretization. Practically, we can use $\rho$RK-DEIS with Eq. (16) and Prop 3 to accelerate likelihood evaluation. We find NLL evaluation based on RK can converge with 36 NFE with 3 order Kutta solver, which reaches 3.16 bits/dim compared with 3.15 bits/dim for RK45 (Song et al., 2020b) and achieves around 4 times acceleration.

2. Q — *Can the proposed method further be accelerated by designing an adaptive step size solver?*

   A — The proposed $\rho$RK-DEIS can be combined with out-of-shelf adaptive step size solvers. However, we find that most ODE trajectories resulting from various starting points share similar patterns in curvature, and a tuned fixed step size works efficiently. Most existing adaptive step size strategies have some probability of getting rejected for the proposed step size, which will waste the NFE budget. Take the example of RK45, one rejection will waste 5 NFE, which is unacceptable when we try to generate samples in 10 NFE or even fewer steps.

3. Q — *The proposed AB-DEIS and iPNDM use lower-order multistep solvers for computing the initial solution. Do they have a convergence guarantee?*

   A — We use lower-order multistep for the first few steps to save computational costs. The strategy can help us achieve similar sampling quality with less NFE as we show in Tabs. 4 and 5, which aligns with our goal of sampling with small NFE. Moreover, lower order Adams-Bashforth methods also enjoy a convergence guarantee, albeit with a slower rate.

4. Q — *How is DEIS compared with the ODE sampling algorithm in Karras et al. (2022)?*

A — We note Karras et al. (2022) is a concurrent work that introduces a second-order Heun method in a rescaled ODE. The algorithm is a special case of $\rho$RK-DEIS with the second-order Heun RK method. Below we show the equivalence. As the two works use different sets of notations, we use blue for notations from Karras et al. (2022) and orange for our notations.

Karras et al. (2022, Algorithm 1) investigates the diffusion model with forward process $\boldsymbol{x}_t \sim \mathcal{N}(s(t)\boldsymbol{x}_0, \sigma(t)^2)$, where $s(t)$ is a scaling factor and $\sigma(t)^2$ represents the variance. Karras et al. (2022, Sec 4) suggests the schedule $s(t) = 1, \sigma(t) = t$, which has the diffusion ODE

$$\frac{d\boldsymbol{x}}{dt} = \frac{\boldsymbol{x} - D(\boldsymbol{x}, t)}{t}, \tag{18}$$

where $D(\boldsymbol{x}, t)$ is trained to predict clean data given noise data $\boldsymbol{x}$ at time $t$. They employ the second-order Huen method to solve Eq. (18). Additionally, they show all isotropic diffusion models with arbitrary $s(t), \sigma(t)$ can be transformed into the suggested diffusion model with parameter schedule $s(t) = 1, \sigma(t) = t$ by proper rescaling. The rescaling in Karras et al. (2022) is equivalent to change-of-variables we introduce in Sec. 4, and Eq. (18) is the simplified ODE Eq. (17) we used that takes into account the analytical form of $\Psi, \boldsymbol{G}_t, \boldsymbol{L}_t$.

To further illustrate the point, consider the example with the popular VPSDE in Prop 3. In this case, the $\rho$RK-DEIS uses the time rescaling $\rho(t) = \sqrt{\frac{1-\alpha_t}{\alpha_t}}$ and the state rescaling $\hat{\boldsymbol{y}}_t = \sqrt{\frac{1}{\alpha_t}}\hat{\boldsymbol{x}}_t$ (note $\alpha_0 = 1$ in VPSDE). The forward process for $\hat{\boldsymbol{y}}_\rho$ becomes

$$\hat{\boldsymbol{y}}_\rho = \hat{\boldsymbol{y}}_{t(\rho)} \sim \mathcal{N}(\hat{\boldsymbol{x}}_0, \frac{1-\alpha_t}{\alpha_t}) = \mathcal{N}(\hat{\boldsymbol{y}}_0, \rho^2), \tag{19}$$

where $t(\rho)$ is the inverse function of $\rho(t)$ and the last equality holds due to $\hat{\boldsymbol{x}}_0 = \hat{\boldsymbol{y}}_0$. Comparing Eq. (19) and the parameter schedule $s(t) = 1, \sigma(t) = t$ in Karras et al. (2022), we conclude that $\hat{\boldsymbol{y}}_\rho$ is equivalent to $\boldsymbol{x}_t$ and $\rho$ is the same as $t$. Moreover, $\frac{\boldsymbol{x} - D(\boldsymbol{x},t)}{t}$ is equivalent to $\epsilon_\theta(\sqrt{\frac{\alpha_{\beta^{-1}(\rho)}}{\alpha_0}}\hat{\boldsymbol{y}}, \beta^{-1}(\rho))$ since both predict added white noise from noised data.

In summary, Karras et al. (2022, Algorithm 1) is a special case of $\rho$RK-DEIS, which can be obtained by employing second-order Heun method in Eq. (17). We include the empirical comparison between other DEIS algorithms and Karras et al. (2022, Algorithm 1), which we denote as $\rho$2Heun. We find with relatively large NFE, third-order Kutta is better than second-order Heun. And $t$AB-DEIS outperforms $\rho$RK-DEIS when NFE is small.

### 5. Q — *How is DEIS compared with sampling algorithm in Lu et al. (2022)?*

A — We note DPM-Solver (Lu et al., 2022) is a concurrent work and it also uses the exponential integrator to reduce discretization error during sampling. Both start with the exact ODE solution but are different at discretization methods for nonlinear score parts. Below we show the connections and differences. As the two works use different sets of notations, we use cyan for notations from Lu et al. (2022) and orange for our notations.

**Exact ODE solution** Lu et al. (2022) investigate diffusion model with forward noising $\boldsymbol{x}_t \sim \mathcal{N}(\alpha_t \boldsymbol{x}_0, \sigma_t^2)$. Lu et al. (2022, Proposition 3.1) propose the exact solution of ODE of $\boldsymbol{x}_t$ given initial value $\boldsymbol{x}_s$ at time $s \geq 0$

$$\boldsymbol{x}_t = \frac{\alpha_t}{\alpha_s}\boldsymbol{x}_s - \alpha_t \int_{\lambda_s}^{\lambda_t} e^{-\lambda}\hat{\epsilon}_\theta(\boldsymbol{x}_\lambda, \lambda)d\lambda, \tag{20}$$

where $\lambda := \log\frac{\alpha_t}{\sigma_t}$ is known as one half of log-SNR (*a.k.a. signal-to-noise-ratio*) and $\hat{\epsilon}_\theta(\boldsymbol{x}_\lambda, \lambda) = \epsilon_\theta(\boldsymbol{x}_t, t)$ with corresponding $t$ given $\lambda$. Similar to exponential Runge-Kutta method (Hochbruck & Ostermann, 2010), Lu et al. (2022) approximate $\int_{\lambda_s}^{\lambda_t} e^{-\lambda}\epsilon_\theta(\boldsymbol{x}_\lambda, \lambda)d\lambda$ based on Taylor expansion and propose DPM-Solvers.

Eq. (20) shares a lot of similarities with $\rho$RK-DEIS. Specifically, $\rho(t) = e^{-\lambda(t)}$ since $\rho = \sqrt{\frac{1-\alpha_t}{\alpha_t}}$, $\sqrt{\alpha_t} = \alpha_t$, and $\sqrt{1-\alpha_t} = \sigma_t$ in VPSDE. Similar to Eq. (20), the exact solution

in Eq. (17) follows

$$\boldsymbol{x}_t = \sqrt{\frac{\alpha_t}{\alpha_s}}\boldsymbol{x}_s + \sqrt{\alpha_t}\int_{\rho_s}^{\rho_t}\hat{\epsilon}(\boldsymbol{x}_\rho, \rho)d\rho, \qquad (21)$$

where $\hat{\epsilon}_\theta(\boldsymbol{x}_\rho, \rho) = \epsilon_\theta(\boldsymbol{x}_t, t)$ with corresponding $t$ given $\rho$. $\rho$RK-DEIS employs out-of-shelf Runge-Kutta solvers for $\int_{\rho_s}^{\rho_t}\hat{\epsilon}(\boldsymbol{x}_\rho, \rho)d\rho$.

**An example of DPM-Solver2** To illustrate the connection and difference more clearly, we consider DPM-Solver-2 and $\rho$RK-DEIS with the standard middle point solver and compare their update schemes. To compare these two algorithms, we first introduce a function $\mathcal{F}_{\text{DDIM}}$ inspired by DDIM. In $\rho$RK-DEIS and DPM-Solver, $\mathcal{F}_{\text{DDIM}}$ is defined as

$$\mathcal{F}_{\text{DDIM}}(\boldsymbol{x}, \boldsymbol{g}, s, t) = \sqrt{\frac{\alpha_t}{\alpha_s}}\boldsymbol{x}_s + [\sqrt{1-\alpha_t} - \sqrt{\frac{\alpha_t}{\alpha_s}}\sqrt{1-\alpha_s}]\boldsymbol{g} \qquad (22)$$

$$\mathcal{F}_{\text{DDIM}}(\boldsymbol{x}, \boldsymbol{g}, s, t) = \frac{\alpha_t}{\alpha_s}\boldsymbol{x}_s + [\sigma_t(e^h - 1)]\boldsymbol{g}, \quad \text{where} \quad h = \lambda_t - \lambda_s \qquad (23)$$

respectively.

With $\mathcal{F}_{\text{DDIM}}$, we can reformulate update schemes of DPM-Solver2 and $\rho$RK-DEIS with midpoint solver into Algo 2 and 3. The two algorithms are only different in the choice of midpoint $s_i$ and $s_i$. In particular, $s_i = \sqrt{\rho_i\rho_{i+1}}$.

**Connection with Runge-Kutta** Though both algorithms are inspired by EI methods and Runge-Kutta, they are actually different even when there is no semi-linear structure in diffusion flow ODE. Let us consider VESDE introduced in Karras et al. (2022) where $\alpha_t = 1, \sigma_t = t$. The VESDE has a simple ODE formulation,

$$d\boldsymbol{x} = \epsilon_\theta(\boldsymbol{x}, t)dt. \qquad (24)$$

Eq. (24) does not have a semi-linear structure. In this case, $\rho$RK-DEIS reduces to standard Runge-Kutta methods and has convergence order $\mathcal{O}(\Delta t^\kappa)$ for $\kappa$-order RK methods. The DPM-solver uses the parametrization $\lambda = -\log(t)$, and is different from standard Runge Kutta and reformulate Eq. (24) as

$$d\boldsymbol{x} = -e^\lambda\epsilon_\theta(\boldsymbol{x}, t_\lambda(\lambda))d\lambda. \qquad (25)$$

For $\kappa$ order DPM-Solver, it has convergence order $\mathcal{O}(\Delta\lambda^\kappa)$ under certain assumptions stated in Lu et al. (2022).

**Empirical comparison** We compare DPM-Solver2, DPM-Solver3, $t$AB-DEIS, and $\rho$RK-DEIS on $64 \times 64$ class-conditioned ImageNet. We observe $t$AB-DEIS has the best sample quality most of time. We believe it is because multistep is better than single-step methods when we have a limited NFEs e.g., 6. DPM-Solvers are better than $\rho$RK-DEIS in small NFE regions and the difference shrinks fastly as we increase sampling steps. We hypothesize that this is because DPM-Solvers are tailored for sampling with small NFEs. However, $t$RK-DEIS has a slightly better FID when NFE is relatively large, although the difference in performance is small. The observation aligns with our experiments in CIFAR10, third order $\rho$RK-DEIS achieves 2.56 with 51 NFE while the third order DPM-Solver achieves 2.65 with 48 NFE (Lu et al., 2022). We include more visual comparison in Figs. 8 and 9.

---

**Algorithm 2** DPM-Solver-2

1: **Input:** $\boldsymbol{x}_i, t_i, t_{i-1}$ and corresponding $\lambda_i, \lambda_{i-1}$
2: **Output:** $\boldsymbol{x}_{i-1}$
3: $s_i = t_\lambda(\frac{\lambda_i + \lambda_{i-1}}{2})$
4: $\boldsymbol{g} = \epsilon_\theta(\boldsymbol{x}_i, t_i)$
5: $\boldsymbol{u}_i = \mathcal{F}_{\text{DDIM}}(\boldsymbol{x}_i, \boldsymbol{g}, t_i, s_i)$
6: $\boldsymbol{g} = \epsilon_\theta(\boldsymbol{u}_i, s_i)$
7: $\boldsymbol{x}_{i-1} = \mathcal{F}_{\text{DDIM}}(\boldsymbol{u}_i, \boldsymbol{g}, s_i, t_{i-1})$

**Algorithm 3** $\rho$RK-DEIS with midpoint solver

1: **Input:** $\boldsymbol{x}_i, t_i, t_{i-1}$ and corresponding $\rho_i, \rho_{i-1}$
2: **Output:** $\boldsymbol{x}_{i-1}$
3: $s_i = t_\rho(\frac{\rho_i + \rho_{i-1}}{2})$
4: $\boldsymbol{g} = \epsilon_\theta(\boldsymbol{x}_i, t_i)$
5: $\boldsymbol{u}_i = \mathcal{F}_{\text{DDIM}}(\boldsymbol{x}_i, \boldsymbol{g}, t_i, s_i)$
6: $\boldsymbol{g} = \epsilon_\theta(\boldsymbol{u}_i, s_i)$
7: $\boldsymbol{x}_{i-1} = \mathcal{F}_{\text{DDIM}}(\boldsymbol{u}_i, \boldsymbol{g}, s_i, t_{i-1})$

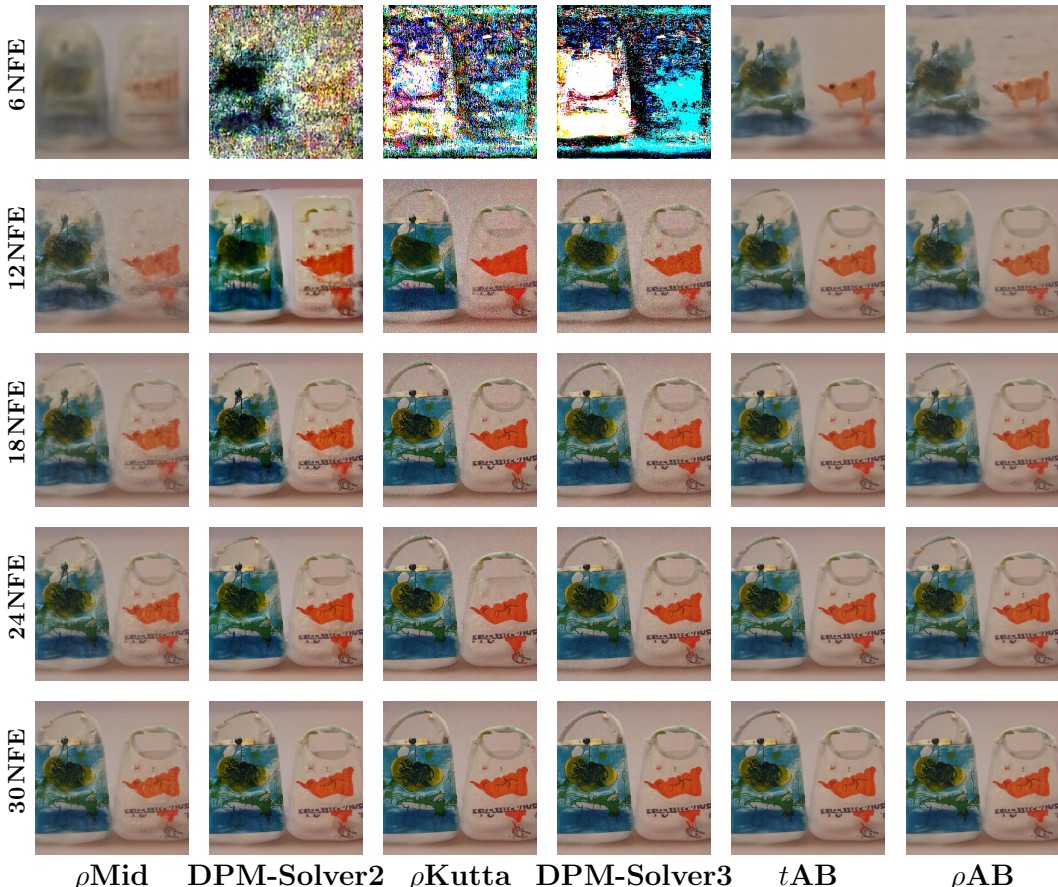

Figure 8: DPM-Solver *v.s.* DEIS with unconditional $256 \times 256$ ImageNet pretrained model (Dhari-wal & Nichol, 2021). (Zoom in to see details)

| | 10 | 12 | 14 | 16 | 18 | 20 | 30 | 50 |
|---|---|---|---|---|---|---|---|---|
| $t$AB | **6.65** | **3.99** | **3.67** | **3.49** | **3.21** | **3.10** | 2.81 | 2.69 |
| $\rho$AB | 9.28 | 6.46 | 5.02 | 4.34 | 3.74 | 3.39 | **2.87** | **2.66** |
| DPM-Solver2 | **7.93** | **5.36** | **4.46** | **3.89** | **3.63** | **3.42** | 3.00 | 2.82 |
| $\rho$Mid | 9.12 | 6.78 | 5.27 | 4.52 | 4.00 | 3.66 | **2.99** | **2.81** |
| DPM-Solver3 | | **5.02** | **3.62**$^+$ | | **3.18** | **3.06**$^+$ | 2.84 | 2.72$^+$ |
| $\rho$Kutta | | 13.12 | 5.18$^+$ | | 3.63 | 3.16$^+$ | **2.82** | **2.71**$^+$ |

Table 3: Comparison between DEIS and DPM-Solver Lu et al. (2022) on class-conditioned $64 \times 64$ ImageNet. The upper right $+$ indicates one extra NFE used.

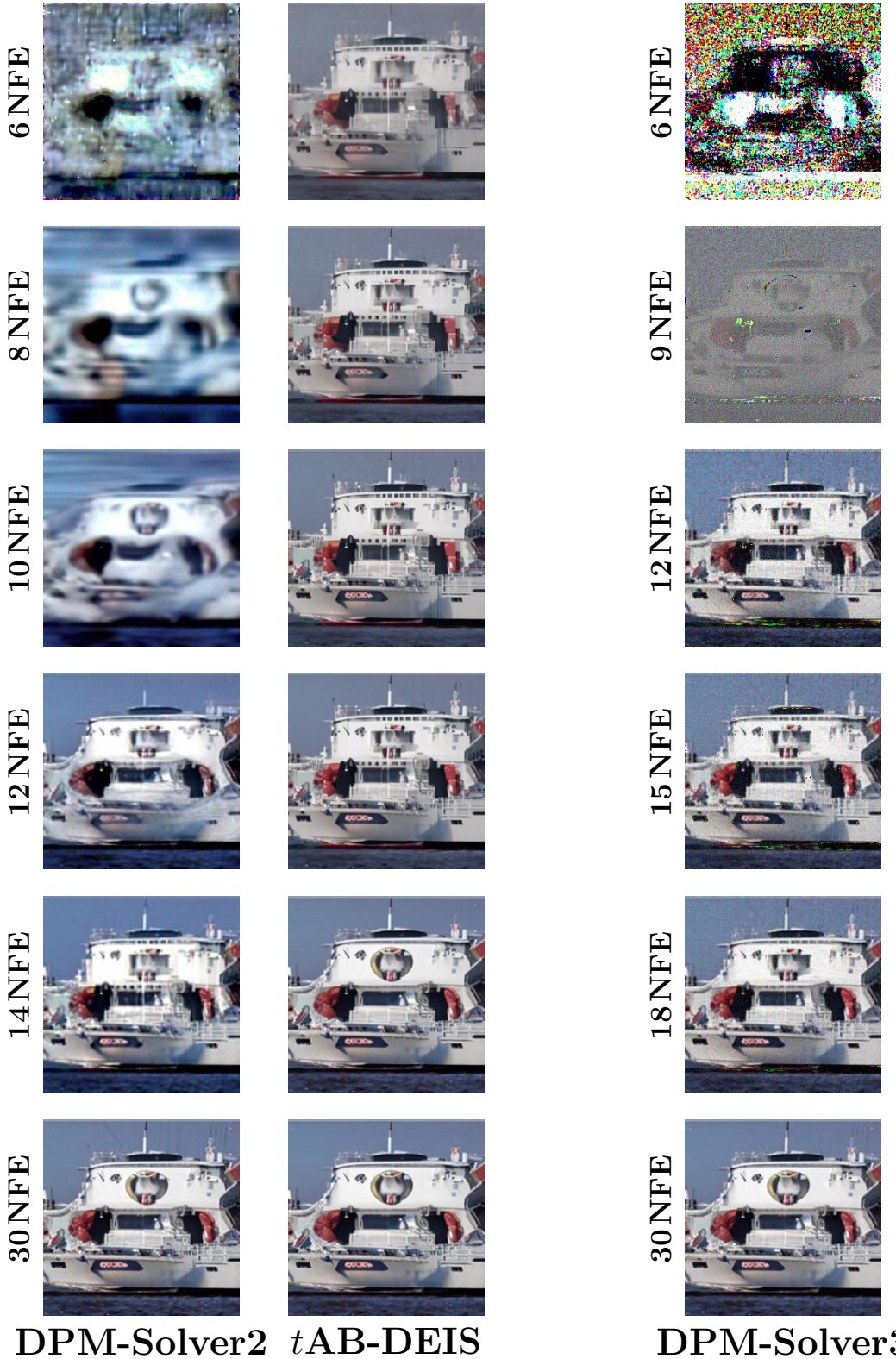

Figure 9: DPM-Solver *v.s.* DEIS with unconditional $256 \times 256$ ImageNet pretrained model (Dhariwal & Nichol, 2021). (Zoom in to see details)

6. Q — *The ODE solvers are sensitive to step size choice. Different works suggest different time discretization (Lu et al., 2022; Karras et al., 2022; Song et al., 2020a). How do compared algorithm and DEIS perform under different step size scheduling?*

A — The comparison given the same time discretization is included in App. H.3. We find different algorithms may prefer different time discretization. We provide a comparison for different sampling algorithms under their best time scheduling in Tab. 2. In most cases especially low NFE region, we find $t$AB-DEIS performs better than other approaches.

7. Q — *Can DEIS be generalized to accelerate SDE sampling for diffusion models?*

A — Some techniques developed in DEIS, such as better score parameterization and analytic treatment of linear term, can be applied to SDE counterparts. However, SDE is more difficult to accelerate compared with ODE. We include more discussions in App. C.

## C  DISCRETIZATION ERROR OF SDE SAMPLING

In this section, we consider the problem of solving the SDE Eq. (4) with $\lambda > 0$. As shown in Prop 1, this would also lead to a sampling scheme from DMs. The exact solution to Eq. (4) satisfies

$$\hat{\boldsymbol{x}}_t = \underbrace{\Psi(t,s)\hat{\boldsymbol{x}}_s}_{\text{Linear term}} + \underbrace{\int_s^t \Psi(t,\tau)\overbrace{\frac{1+\lambda^2}{2}}^{\text{Weight}}\boldsymbol{G}_\tau\boldsymbol{G}_\tau^T\boldsymbol{L}_\tau^{-T}\,\epsilon_\theta(\hat{\boldsymbol{x}}_\tau,\tau)d\tau}_{\text{Nonlinear term}} + \int_s^t \lambda\underbrace{\Psi(t,\tau)\boldsymbol{G}_\tau d\boldsymbol{w}}_{\text{Noise term}}, \qquad (26)$$

where $\Psi$ is as before. The goal is to approximate Eq. (26) through discretization. Interestingly, the stochastic DDIM (Song et al., 2020a) turns out to be a numerical solver for Eq. (26) as follows (Proof in App. G).

**Proposition 4.** *For the VPSDE, the stochastic DDIM is a discretization scheme of Eq. (26).*

How do we discretize Eq. (26) for a general SDE Eq. (4)? One strategy is to follow what we did for the ODE ($\lambda = 0$) in Sec. 3.2 and approximate $\epsilon_\theta(\hat{\boldsymbol{x}}_\tau, \tau)$ by a polynomial. However, we found this strategy does not work well in practice. We believe it is due to several possible reasons as follows. We do not pursue the discretization of the SDE Eq. (4) further in this paper and leave it for future.

**Nonlinear weight and discretization error**. In Eq. (26), the linear and noise terms can be calculated exactly without discretizaiton error. Thus, only the nonlinear term $\epsilon_\theta$ can induce error in the EI method. Compared with Eq. (11), Eq. (26) has a larger weight for the nonlinearity term as $\lambda > 0$ and is therefore more likely to cause larger errors. From this perspective, the ODE with $\lambda = 0$ is the best option since it minimizes the weight of nonlinear term. In Song et al. (2020a), the authors also observed that the deterministic DDIM outperforms stochastic DDIM. Such observation is consistent with our analysis. Besides, we notice that the nonlinear weight in VPSDE is significantly smaller than that in VESDE, which implies VPSDE has smaller discretization error. Indeed, empirically, VPSDE has much better sampling performance when $N$ is small. **Additional noise**. Compared with Eq. (11) for ODEs, Eq. (26) injects additional noise to the state when it is simulated backward. Thus, to generate new samples by denoising, the score model needs to not only remove noise in $\hat{\boldsymbol{x}}_{t_N}$, but also remove this injected noise. For this reason, a better approximation of $\epsilon_\theta$ may be needed.

## D  PROOF OF PROP 1

The proof is inspired by (Zhang & Chen, 2021). We show that the marginal distribution induced by Eq. (4) does not depend on the choice of $\lambda$ and equals the marginal distribution induced by Eq. (2) when the score model is perfect.

Consider the distribution $q$ induced by the SDE

$$d\boldsymbol{x} = [\boldsymbol{F}_t\boldsymbol{x} - \frac{1+\lambda^2}{2}\boldsymbol{G}_t\boldsymbol{G}_t^T\nabla\log q_t(\boldsymbol{x})]dt + \lambda\boldsymbol{G}_t d\boldsymbol{w}. \qquad (27)$$

Eq. (27) is simulated from $t = T$ to $t = 0$. According to the Fokker-Planck-Kolmogorov (FPK) Equation, $q$ solves the partial differential equation

$$\frac{\partial q_t(\boldsymbol{x})}{\partial t} = -\nabla \cdot \{[\boldsymbol{F}_t \boldsymbol{x} - \frac{1+\lambda^2}{2} \boldsymbol{G}_t \boldsymbol{G}_t^T \nabla \log q_t(\boldsymbol{x})]q_t(\boldsymbol{x})\} - \frac{\lambda^2}{2}\langle \boldsymbol{G}_t \boldsymbol{G}_t^T, \frac{\partial^2}{\partial x_i \partial x_j}q_t(\boldsymbol{x})\rangle$$

$$= -\nabla \cdot \{[\boldsymbol{F}_t \boldsymbol{x} - \frac{1}{2}\boldsymbol{G}_t \boldsymbol{G}_t^T \nabla \log q_t(\boldsymbol{x})]q_t(\boldsymbol{x})\} + \nabla \cdot \{[\frac{\lambda^2}{2}\boldsymbol{G}_t \boldsymbol{G}_t^T \nabla \log q_t(\boldsymbol{x})]q_t(\boldsymbol{x})\} -$$

$$\frac{\lambda^2}{2}\langle \boldsymbol{G}_t \boldsymbol{G}_t^T, \frac{\partial^2}{\partial x_i \partial x_j}q_t(\boldsymbol{x})\rangle,$$

where $\nabla\cdot$ denotes the divergence operator. Since

$$\nabla \cdot \{[\frac{\lambda^2}{2}\boldsymbol{G}_t \boldsymbol{G}_t^T \nabla \log q_t(\boldsymbol{x})]q_t(\boldsymbol{x})\} = \nabla \cdot [\frac{\lambda^2}{2}\boldsymbol{G}_t \boldsymbol{G}_t^T \nabla q_t(\boldsymbol{x})] = \langle \frac{\lambda^2}{2}\boldsymbol{G}_t \boldsymbol{G}_t^T, \frac{\partial^2}{\partial x_i \partial x_j}q_t(\boldsymbol{x})\rangle, \quad (28)$$

we obtain

$$\frac{\partial q_t(\boldsymbol{x})}{\partial t} = -\nabla \cdot \{[\boldsymbol{F}_t \boldsymbol{x} - \frac{1}{2}\boldsymbol{G}_t \boldsymbol{G}_t^T \nabla \log q_t(\boldsymbol{x})]q_t(\boldsymbol{x})\}. \quad (29)$$

Eq. (29) shows that the above partial differential equation does not depend on $\lambda$. Thus, the marginal distribution of Eq. (27) is independent of the value of $\lambda$.

## E    PROOF OF PROP 2

Thanks to , A straightforward calculation based on Eq. (6) gives that $\Psi(t, s)$ for the VPSDE is

$$\Psi(t, s) = \sqrt{\frac{\alpha_t}{\alpha_s}}.$$

It follows that

$$\int_s^t \Psi(t, \tau)\frac{1}{2}\boldsymbol{G}_\tau \boldsymbol{G}_\tau^T \boldsymbol{L}_\tau^{-1} d\tau = \int_s^t -\frac{1}{2}\sqrt{\frac{\alpha_t}{\alpha_\tau}}\frac{d \log \alpha_\tau}{d\tau}\frac{1}{\sqrt{1-\alpha_\tau}}d\tau$$

$$= \sqrt{\alpha_t}\int_s^t -\frac{1}{2}\frac{d\alpha_\tau}{\alpha_\tau^{1.5}(1-\alpha_\tau)^{0.5}}$$

$$= \sqrt{\alpha_t}\left.\sqrt{\frac{1-\tau}{\tau}}\right|_{\alpha_s}^{\alpha_t}$$

$$= \sqrt{1-\alpha_t} - \sqrt{\frac{\alpha_t}{\alpha_s}}\sqrt{1-\alpha_s}.$$

Setting $t \leftarrow t - \Delta t, s \leftarrow t$, we write Eq. (11) as

$$\hat{\boldsymbol{x}}_{t-\Delta t} = \sqrt{\frac{\alpha_{t-\Delta t}}{\alpha_t}}\hat{\boldsymbol{x}}_t + [\sqrt{1-\alpha_{t-\Delta t}} - \sqrt{\frac{\alpha_{t-\Delta t}}{\alpha_t}}\sqrt{1-\alpha_t}]\epsilon_\theta(\hat{\boldsymbol{x}}_t, t).$$

## F    PROOF OF PROP 3

We start our proof with Eq. (16). In VPSDE, Eq. (16) reduce to

$$\frac{d\hat{\boldsymbol{y}}}{dt} = -\frac{1}{2}\sqrt{\frac{\alpha_0}{\alpha_t}}\frac{d \log \alpha_t}{dt}\frac{1}{\sqrt{1-\alpha_t}}\epsilon_\theta(\Psi(0, t)\hat{\boldsymbol{y}}, t). \quad (30)$$

Now we consider a rescaled time $\rho$, which satisfies the following equation

$$\frac{d\rho}{dt} = -\frac{1}{2}\sqrt{\frac{\alpha_0}{\alpha_t}}\frac{d \log \alpha_t}{dt}\frac{1}{\sqrt{1-\alpha_t}}. \quad (31)$$

Plugging Eq. (31) into Eq. (30), we reach

$$\frac{d\hat{\boldsymbol{y}}}{d\rho} = \epsilon_\theta(\Psi(0,t)\hat{\boldsymbol{y}},t). \tag{32}$$

In VPSDE, we $\alpha_t$ is a monotonically decreasing function with respect to $t$. Therefore, there exists a bijective mapping between $\rho$ and $t$ based on Eq. (31), which we define as $\beta$ and $\rho = \beta(t)$. Furthermore, we can solve Eq. (31) for $\beta$

$$\beta(t) = \sqrt{\alpha_0}\left(\sqrt{\frac{1-\alpha_t}{\alpha_t}} - \sqrt{\frac{1-\alpha_0}{\alpha_0}}\right). \tag{33}$$

## G    PROOF OF PROP 4

Our derivation uses the notations in (Song et al., 2020a). The DDIM employs the update step

$$\boldsymbol{x}_{t-\Delta t} = \sqrt{\alpha_{t-\Delta t}}\left(\frac{\boldsymbol{x}_t - \sqrt{1-\alpha_t}\epsilon_\theta(\boldsymbol{x}_t,t)}{\sqrt{\alpha_t}}\right) + \sqrt{1 - \alpha_{t-\Delta t} - \eta^2\frac{1-\alpha_{t-\Delta t}}{1-\alpha_t}\left(1 - \frac{\alpha_t}{\alpha_{t-\Delta t}}\right)}\epsilon_\theta(\boldsymbol{x}_t,t)$$
$$+ \eta\sqrt{\frac{1-\alpha_{t-\Delta t}}{1-\alpha_t}\left(1 - \frac{\alpha_t}{\alpha_{t-\Delta t}}\right)}\epsilon_t, \tag{34}$$

where $\eta$ is a hyperparameter and $\eta \in [0,1]$. When $\eta = 0$, Eq. (34) becomes determinstic and reduces to Eq. (12). We show that Eq. (34) is equivalent to Eq. (4) when $\eta = \lambda$ and $\Delta t \to 0$.

By Eq. (34), $\boldsymbol{x}_{t-\Delta t} \sim \mathcal{N}(\mu_\eta, \sigma_\eta^2 \boldsymbol{I})$, where

$$\mu_\eta = \sqrt{\alpha_{t-\Delta t}}\left(\frac{\boldsymbol{x}_t - \sqrt{1-\alpha_t}\epsilon_\theta(\boldsymbol{x}_t,t)}{\sqrt{\alpha_t}}\right) + \sqrt{1 - \alpha_{t-\Delta t} - \eta^2\frac{1-\alpha_{t-\Delta t}}{1-\alpha_t}\left(1 - \frac{\alpha_t}{\alpha_{t-\Delta t}}\right)}\epsilon_\theta(\boldsymbol{x}_t,t)$$
$$\sigma_\eta^2 = \eta^2\frac{1-\alpha_{t-\Delta t}}{1-\alpha_t}\left(1 - \frac{\alpha_t}{\alpha_{t-\Delta t}}\right).$$

It follows that

$$\lim_{\Delta t \to 0} \frac{\boldsymbol{x}_t - \mu_\eta}{t - (t-\Delta t)} = \frac{\left(1 - \sqrt{\frac{\alpha_{t-\Delta t}}{\alpha_t}}\right)}{\Delta t}\boldsymbol{x}_t$$
$$+ \frac{\sqrt{\alpha_{t-\Delta t}}\sqrt{\frac{1-\alpha_t}{\alpha_t}} - \sqrt{1-\alpha_{t-\Delta t} - \eta^2\frac{1-\alpha_{t-\Delta t}}{1-\alpha_t}\left(1 - \frac{\alpha_t}{\alpha_{t-\Delta t}}\right)}}{\Delta t}\epsilon_\theta(\boldsymbol{x}_t,t)$$
$$= \frac{1}{2}\frac{d\log\alpha_t}{dt}\boldsymbol{x} + \frac{1+\eta^2}{2}\frac{d\log\alpha_t}{dt}\frac{1}{\sqrt{1-\alpha_t}}\epsilon_\theta(\boldsymbol{x}_t,t)$$
$$= \boldsymbol{F}_t\boldsymbol{x} + \frac{1+\eta^2}{2}\boldsymbol{G}_t\boldsymbol{G}_t^T\boldsymbol{L}_t^{-T}\epsilon_\theta(\boldsymbol{x}_t,t),$$

and

$$\lim_{\Delta t \to 0} \frac{\eta^2\frac{1-\alpha_{t-\Delta t}}{1-\alpha_t}\left(1 - \frac{\alpha_t}{\alpha_{t-\Delta t}}\right)}{dt} = -\eta^2\frac{d\log\alpha_t}{dt} = \eta^2\boldsymbol{G}_t\boldsymbol{G}_t^T.$$

Consequently, the continuous limit of Eq. (34) is

$$d\boldsymbol{x} = \left[\boldsymbol{F}_t\boldsymbol{x} + \frac{1+\eta^2}{2}\boldsymbol{G}_t\boldsymbol{G}_t^T\boldsymbol{L}_t^{-T}\epsilon_\theta(\boldsymbol{x},t)\right]dt + \eta\boldsymbol{G}_t d\boldsymbol{w}, \tag{35}$$

which is exactly Eq. (4) if $\eta = \lambda$.

## H MORE EXPERIMENT DETAILS

### H.1 IMPORTANT TECHNICAL DETAILS AND MODIFICATIONS

- In Sec. 3, the ground-truth solutions $\{\hat{x}_t^*\}$ are approximated by solving ODE with high accuracy solvers and small step size. We empirically find solutions of RK4 converge when step size smaller than $2 \times 10^{-3}$ in VPSDE. We approximated ground-truth solutions by RK4 solutions with step size $1 \times 10^{-3}$.

- It is found that correcting steps and an extra denoising step can improve image quality at additional NFE costs (Song et al., 2020b; Jolicoeur-Martineau et al., 2021). For a fair comparison, we disable the correcting steps, extra denoising step, or other heuristic clipping tricks for all methods and experiments in this work unless stated otherwise.

- Due to numerical issues, we set ending time $t_0$ in DMs during sampling a non-zero number. Song et al. (2020b) suggests $t_0 = 10^{-3}$ for VPSDE and $t_0 = 10^{-5}$ for VESDE. In practice, we find the value of $t_0$ and time scheduling have huge impacts on FIDs. This finding is not new and has been pointed out by existing works (Jolicoeur-Martineau et al., 2021; Kim et al., 2021; Song et al., 2020a). Interestingly, we found different algorithms have different preferences for $t_0$ and time scheduling. We report the best FIDs for each method among different choices of $t_0$ and time scheduling in Tab. 2. We use $t_0$ suggested by the original paper and codebase for different checkpoints and quadratic time scheduling suggested by Song et al. (2020a) unless stated otherwise. We include a comprehensive study about $t_0$ and time scheduling in App. H.3

- Because PNDM needs 12 NFE for the first 3 steps, we compare PNDM only when NFE is great than 12. However, our proposed iPNDM can work when NFE is less than 12.

- We include the comparison against A-DDIM (Bao et al., 2022) with its official checkpoints and implementation in App. H.5.

- We only provide qualitative results for text-to-image experiment with pre-trained model (Ramesh et al., 2022).

- We include proposed $r$-th order iPNDM in App. H.2. We use $r = 3$ by default unless stated otherwise.

### H.2 IMPROVED PNDM

By Eq. (11), PNDM can be viewed as a combination of Exponential Integrator and linear multistep method based on the Euler method. More specifically, it uses a linear combination of multiple score evaluations instead of using only the latest score evaluation. PNDM follows the steps

$$\hat{\epsilon}_t^{(3)} = \frac{1}{24}(55\epsilon_t - 59\epsilon_{t+\Delta t} + 37\epsilon_{t+2\Delta t} - 9\epsilon_{t+3\Delta t}), \tag{36}$$

$$\hat{x}_{t-\Delta t} = \sqrt{\frac{\alpha_{t-\Delta t}}{\alpha_t}}\hat{x}_t + [\sqrt{1 - \alpha_{t-\Delta t}} - \sqrt{\frac{\alpha_{t-\Delta t}}{\alpha_t}}\sqrt{1 - \alpha_t}]\hat{\epsilon}_t^{(4)}, \tag{37}$$

where $\epsilon_t = \epsilon_\theta(\hat{x}_t, t), \epsilon_{t+\Delta t} = \epsilon_\theta(\hat{x}_{t+\Delta t}, t + \Delta t)$. The coefficients in Eq. (36) are derived based on black-box ODE Euler discretization with fixed step size. Similarly, there exist lower order approximations

$$\hat{\epsilon}_t^{(0)} = \epsilon_t \tag{38}$$

$$\hat{\epsilon}_t^{(1)} = \frac{3}{2}\epsilon_t - \frac{1}{2}\epsilon_{t+\Delta t} \tag{39}$$

$$\hat{\epsilon}_t^{(2)} = \frac{1}{12}(23\epsilon_t - 16\epsilon_{t+\Delta t} + 5\epsilon_{t+2\Delta t}). \tag{40}$$

Originally, PNDM uses Runge-Kutta for warming start and costs 4 score network evaluation for each of the first 3 steps. To reduce the NFE in sampling, the improved PNDM (iPNDM) uses lower order multistep for warming start. We summarize iPNDM in Algo 4. We include a comparison with $t$AB-DEIS in Tabs. 4 and 5, we adapt uniform step size for $t$AB-DEIS when NFE=50 in CIFAR10 as we find its performance is slightly better than the quadratic one.

---

**Algorithm 4** Improved PNDM (iPNDM)

---

**Input**: $\{t_i\}_{i=0}^N, t_i = i\Delta t$, order $r$
**Instantiate**: $\boldsymbol{x}_{t_N}$, Empty $\epsilon$-buffer
**for** $i$ in $N, N-1, \cdots, 1$ **do**
   $j = \min(N - i + 1, r)$
   $\epsilon$-buffer.append($\epsilon_\theta(\hat{\boldsymbol{x}}_{t_i}, t_i)$)
   Simulate $\hat{\epsilon}_{t_i}^{(j)}$ based on $j$ and $\epsilon$-buffer
   $\hat{\boldsymbol{x}}_{t_{i-1}} \leftarrow$ Simulate Eq. (37) with $\hat{\boldsymbol{x}}_{t_i}$ and $\hat{\epsilon}_{t_i}^{(j)}$
**end for**

---

| FID\NFE | | | | |
|---|---|---|---|---|
| Method | 5 | 10 | 20 | 50 |
| PNDM | - | - | 6.42 | 3.03 |
| iPNDM | 70.07 | 9.36 | 4.21 | 3.00 |
| DDIM | 30.64 | 11.71 | 6.12 | 4.25 |
| $t$AB1 | 20.01 | 6.09 | 3.81 | 3.32 |
| $t$AB2 | 16.53 | 4.57 | 3.41 | 3.09 |
| $t$AB3 | **16.10** | **4.17** | **3.33** | **2.99** |

Table 4: PNDM and iPNDM on CIFAR10

### H.3 IMPACT OF $t_0$ AND TIME SCHEDULING ON FIDS

**Ingredient 4: Optimizing time discretization.** From Fig. 4 we observe that the approximation error is not uniformly distributed for all $t_0 \leq t \leq t_N$ when uniform discretization over time is used; the error increases as $t$ approaches $0$. This observation implies that, instead of a uniform step size (linear timesteps), a smaller step size should be used for $t$ close to $0$ to improve accuracy. One such option is the quadratic timestep suggested in (Song et al., 2020a) that follows linspace($t_0, \sqrt{t_N}, N + 1)^2$.

To better understand the effects of time discretization, we investigate the difference between the ground truth $\hat{\boldsymbol{x}}_t^*$ and the numerical solution $\hat{\boldsymbol{x}}_t$ with the same boundary value $\hat{\boldsymbol{x}}_T^*$

$$\hat{\boldsymbol{x}}_t^* - \hat{\boldsymbol{x}}_t = \int_T^t \frac{1}{2}\Psi(t, \tau)\boldsymbol{G}_\tau \boldsymbol{G}_\tau^T \boldsymbol{L}_\tau^{-T}\Delta\epsilon_\theta(\tau)d\tau, \quad \Delta\epsilon_\theta(\tau) = \epsilon_\theta(\hat{\boldsymbol{x}}_\tau^*, \tau) - \boldsymbol{P}_r(\tau). \quad (41)$$

Eq. (41) shows that the difference between the solutions $\hat{\boldsymbol{x}}_t^*$ and $\hat{\boldsymbol{x}}_t$ is a weighted sum of $\Delta\epsilon_\theta(\tau)$. We emphasize that Eq. (41) does not only contain the approximation error of $\boldsymbol{P}_r(\tau)$ which we discussed before, but also accumulation error. Indeed, since $\boldsymbol{P}_r(\tau)$ is fitted on the solution $\{\hat{\boldsymbol{x}}_\tau\}$ instead of ground truth trajectory $\{\hat{\boldsymbol{x}}_\tau^*\}$, there exists accumulation error caused by past errors. A good choice of time discretization should balance the approximation error and the accumulation error.

We have two options for time discretization, adaptive step size, and fixed timestamps. There exists one unique ODE for DMs and we find various ODE trajectories share a similar pattern of curva-

| FID\NFE | | | | |
|---|---|---|---|---|
| Method | 5 | 10 | 20 | 50 |
| PNDM | - | - | 7.60 | 3.51 |
| iPNDM | 59.87 | 7.78 | 5.58 | 3.34 |
| 0-DEIS | 30.42 | 13.53 | 6.89 | 4.17 |
| 1-DEIS | 26.65 | 8.81 | 4.33 | 3.19 |
| 2-DEIS | 25.13 | 7.20 | 3.61 | 3.04 |
| 3-DEIS | **25.07** | **6.95** | **3.41** | **2.95** |

Table 5: PNDM and iPNDM on CELEBA

ture empirically. And the cost of rejected steps in adaptive step size solvers is not ignorable when our NFE is small, such as 10 or even 5. Thus, we prefer and explore fixed timestamps in DEIS. We experiment with several popular options for time discretization (Salimans & Ho, 2022; Song et al., 2020a) in H.3. Surprisingly, given the different budgets of NFE, we find various samplers have different preferences for timesteps. How to design time discretization in a symmetrical approach is an interesting problem; we leave it for future research. In Fig. 5, we show the effects of each ingredient we introduce. With Exponential Integrator, other ingredients can consistently improve sampling quality in terms of FID. Compared with other sampling algorithms, DEIS enjoys significant acceleration.

We present a study about sampling with difference $t_0$ and time scheduling based VPSDE. We consider two choices of $t_0$ $(10^{-3}, 10^{-4})$ and three choices for time scheduling. The first time scheduling follows the power function in $t$

$$t_i = (\frac{N-i}{N} t_0^{\frac{1}{\kappa}} + \frac{i}{N} t_N^{\frac{1}{\kappa}})^{\kappa}, \tag{42}$$

the second time scheduling follows power function in $\rho$

$$\rho_i = (\frac{N-i}{N} \rho_0^{\frac{1}{\kappa}} + \frac{i}{N} \rho_N^{\frac{1}{\kappa}})^{\kappa}, \tag{43}$$

and the last time scheduling follows a uniform step in $\log \rho$ space

$$\log \rho_i = \frac{N-i}{N} \log \rho_0 + \frac{i}{N} \log \rho_N. \tag{44}$$

We include the comparison between different $t_0$ and time scheduling in Tabs. 6 to 8. We notice $t_0$ has a huge influence on image FIDs, which is also noticed and investigated across different studies (Kim et al., 2021; Dockhorn et al., 2021). Among various scheduling, we observe $t$AB-DEIS has obvious advantages when NFE is small and $\rho$RK-DEIS is competitive when we NFE is relatively large.

## H.4 MORE ABALATION STUDY

We include more quantitative comparisons of the introduced ingredients in Tab. 9 for Fig. 5. Since ingredients $\epsilon_\theta$-based parameterization and polynomial extrapolation are only compatible with the exponential integrator, we cannot combine them with the Euler method. We also provide performance when applying quadratic timestamp scheduling to Euler Tab. 10 directly. We find sampling with small NFE and large NFE have different preferences for time schedules.

We also report the performance of the RK45 ODE solver for VPSDE on CIFAR10 in Tab. 11 [3]. As a popular and well-developed ODE solver, RK45 has decent sampling performance when NFE $\geq 50$. However, the sampling quality with limited NFE is not satisfying. Such results are within expectation as RK45 does not take advantage of the structure information of diffusion models. The overall performance of RK45 solver is worse than iPNDM and DEIS when NFE is small.

## H.5 COMPARISON WITH ANALYTIC-DDIM (A-DDIM) (BAO ET AL., 2022)

We also compare our algorithm with Analytic-DDIM (A-DDIM) in terms of fast sampling performance. We failed to reproduce the significant improvements claimed in (Bao et al., 2022) in our default CIFAR10 checkpoint. There could be two factors that contribute to this. First, we use a score network trained with continuous time loss objective and different weights (Song et al., 2020b). However, Analytic-DDIM is proposed for DDPM with discrete times and finite timestamps. Second, some tricks have huge impacts on the sampling quality in A-DDIM. For instance, A-DDIM heavily depends on clipping value in the last few steps (Bao et al., 2022). A-DDIM does not provide high-quality samples without proper clipping when NFE is low.

To compare with A-DDIM, we conduct another experiment with checkpoints provided by (Bao et al., 2022) and integrate iPNDM and DEIS into the provided codebase; the results are shown in Tab. 12. We use piecewise linear function to fit discrete SDE coefficients in (Bao et al., 2022) for DEIS. Without any ad-hoc tricks, the plugin-and-play iPNDM is comparable or even slightly better than A-DDIM when the NFE budget is small, and DEIS is better than both of them.

---

[3]We use $scipy.integrate.solve\_ivp$ and tune tolerance to get different performances on different NFE. We find different combinations of absolute tolerance and relative tolerance may result in the same NFE but different FID. We report the best FID in that case.

| | FID for various DEIS with $\kappa = 1$ in Eq. (42) | | | | | | | | | |
|---|---|---|---|---|---|---|---|---|---|---|
| NFE | DDIM | $\rho$2Heun | $\rho$3Kutta | $\rho$4RK | $\rho$AB1 | $\rho$AB2 | $\rho$AB3 | $t$AB1 | $t$AB2 | $t$AB3 |
| 5 | 47.59 | $207^{+1}$ | $238^{+1}$ | $212^{+3}$ | 35.14 | 32.51 | 32.02 | 25.99 | **25.06** | 44.29 |
| 10 | 16.60 | 84.55 | $66.81^{+2}$ | $78.57^{+2}$ | 10.47 | 8.85 | 8.18 | 9.51 | 7.71 | **7.18** |
| 15 | 10.39 | $46.36^{+1}$ | 47.45 | $41.27^{+1}$ | 6.69 | 5.70 | 5.24 | 6.47 | 5.51 | **5.01** |
| 20 | 7.93 | 34.87 | $28.35^{+1}$ | 27.21 | 5.27 | 4.56 | 4.24 | 5.20 | 4.50 | **4.14** |
| 50 | 4.36 | 11.58 | $7.00^{+1}$ | $7.48^{+2}$ | 3.32 | 3.08 | **2.99** | 3.32 | 3.09 | **2.99** |
| | FID for various DEIS with $\kappa = 2$ in Eq. (42) | | | | | | | | | |
| NFE | DDIM | $\rho$2Heun | $\rho$3Kutta | $\rho$4RK | $\rho$AB1 | $\rho$AB2 | $\rho$AB3 | $t$AB1 | $t$AB2 | $t$AB3 |
| 5 | 30.64 | $256^{+1}$ | $357^{+1}$ | $342^{+3}$ | 24.58 | 23.60 | 23.48 | 20.01 | 16.52 | **16.10** |
| 10 | 11.71 | 56.62 | $56.51^{+2}$ | $103^{+2}$ | 7.56 | 6.72 | 6.50 | 6.09 | 4.57 | **4.17** |
| 15 | 7.67 | $10.62^{+1}$ | 14.96 | $36.15^{+1}$ | 4.93 | 4.40 | 4.26 | 4.29 | 3.57 | **3.37** |
| 20 | 6.11 | 6.33 | $4.74^{+1}$ | 12.81 | 4.16 | 3.84 | 3.77 | 3.81 | 3.41 | **3.33** |
| 50 | 4.24 | 3.88 | $3.75^{+1}$ | $3.78^{+2}$ | 3.70 | 3.68 | 3.69 | 3.62 | **3.61** | 3.36 |
| | FID for various DEIS with $\kappa = 3$ in Eq. (42) | | | | | | | | | |
| NFE | DDIM | $\rho$2Heun | $\rho$3Kutta | $\rho$4RK | $\rho$AB1 | $\rho$AB2 | $\rho$AB3 | $t$AB1 | $t$AB2 | $t$AB3 |
| 5 | 34.07 | $356^{+1}$ | $388^{+1}$ | $377^{+3}$ | 29.80 | 29.35 | 29.38 | 24.87 | 22.57 | **22.06** |
| 10 | 14.59 | 115 | $171^{+2}$ | $267^{+2}$ | 10.73 | 10.16 | 10.11 | 8.11 | 6.36 | **5.97** |
| 15 | 9.22 | $32.94^{+1}$ | 77.44 | $103^{+1}$ | 6.45 | 6.03 | 5.98 | 5.21 | 4.26 | **4.05** |
| 20 | 7.27 | 13.06 | $11.55^{+1}$ | 50.56 | 5.17 | 4.83 | 4.78 | 4.45 | 3.88 | **3.75** |
| 50 | 4.64 | 3.76 | $\mathbf{3.68}^{+1}$ | $3.74^{+2}$ | 3.92 | 3.82 | 3.79 | 3.81 | 3.72 | 3.71 |
| | FID for various DEIS with Eq. (44) | | | | | | | | | |
| NFE | DDIM | $\rho$2Heun | $\rho$3Kutta | $\rho$4RK | $\rho$AB1 | $\rho$AB2 | $\rho$AB3 | $t$AB1 | $t$AB2 | $t$AB3 |
| 5 | 54.58 | $216^{+1}$ | $335^{+1}$ | $313^{+3}$ | 49.25 | 48.56 | 48.47 | 37.99 | 28.45 | **26.11** |
| 10 | 20.03 | 14.72 | $13.19^{+2}$ | $28.65^{+2}$ | 14.05 | 12.63 | 12.18 | 10.36 | 7.03 | **5.71** |
| 15 | 11.99 | $5.03^{+1}$ | 5.88 | $6.88^{+1}$ | 7.72 | 6.67 | 6.29 | 6.22 | 4.69 | **4.13** |
| 20 | 8.92 | 4.12 | $3.97^{+1}$ | 4.14 | 5.79 | 5.05 | 4.78 | 4.97 | 4.10 | **3.80** |
| 50 | 5.05 | **3.67** | $3.75^{+1}$ | $3.73^{+2}$ | 4.01 | 3.84 | 3.79 | 3.89 | 3.74 | 3.72 |

Table 6: DEIS for VPSDE on CIFAR10 with $t_0 = 10^{-3}$.

### H.6 Sampling quality on ImageNet $32 \times 32$

We conduct experiments on ImageNet $32 \times 32$ with pre-trained VPSDE model provided in (Song et al., 2021a). Again, we observe significant improvement over DDIM and iPNDM methods when the NFE budget is low. Even with 50 NFE, DEIS is able to outperform blackbox ODE solver in terms of sampling quality.

### H.7 Details of experiments on ImageNet $64 \times 64$ and Bedroom $256 \times 256$

We use popular checkpoints from guided-diffusion[4] for our class-conditioned ImageNet $64 \times 64$ and $256 \times 256$ LSUN bedroom experiments. Though the models are trained with discrete time, we simply treat them as continuous diffusion models. Better performance is possible if we have a better time discretization scheme. We adopt time scheduling with $\kappa = 7$ in Eq. (43) suggested by Karras et al. (2022) with $\rho_1 = 0.002, \rho_N = 80.0$, which gives a better empirical performance in class-conditioned ImageNet. We also use Eq. (44) time scheduling suggested by Lu et al. (2022) and $\rho_1 = 0.002, \rho_N = 80.0$. Better sampling quality may be obtained with different time discretization.

### H.8 More results on VPSDE

We include mean and standard deviation for CELEBA in Tab. 14.

---

[4] https://github.com/openai/guided-diffusion

| NFE | DDIM | $\rho$2Heun | $\rho$3Kutta | $\rho$4RK | $\rho$AB1 | $\rho$AB2 | $\rho$AB3 | $t$AB1 | $t$AB2 | $t$AB3 |
|-----|------|-------------|--------------|-----------|-----------|-----------|-----------|--------|--------|--------|
| | | | FID for various DEIS with $\kappa = 1$ in Eq. (42) | | | | | | | |
| 5 | 42.38 | $239^{+1}$ | $232^{+1}$ | $199^{+3}$ | 33.09 | 31.06 | 30.67 | 26.01 | **20.57** | 42.35 |
| 10 | 17.23 | 143 | $95.13^{+2}$ | $130^{+2}$ | 12.44 | 11.04 | 10.41 | 12.01 | 10.57 | **8.04** |
| 15 | 12.06 | $99.76^{+1}$ | 77.37 | $88.56^{+1}$ | 9.12 | 8.25 | 7.79 | 9.08 | 8.20 | **7.50** |
| 20 | 9.71 | 82.89 | $57.54^{+1}$ | 66.61 | 7.57 | 6.89 | 6.50 | 7.60 | 6.90 | **6.45** |
| 50 | 5.76 | 31.56 | $13.10^{+1}$ | $15.73^{+2}$ | 4.64 | 4.25 | **4.06** | 4.67 | 4.28 | 4.10 |
| | | | FID for various DEIS with $\kappa = 2$ in Eq. (42) | | | | | | | |
| 5 | 26.91 | $271^{+1}$ | $362^{+1}$ | $348^{+3}$ | 22.28 | 21.53 | 21.43 | 19.72 | 16.31 | **15.37** |
| 10 | 11.14 | 66.25 | $63.53^{+2}$ | $111^{+2}$ | 7.65 | 6.89 | 6.67 | 6.74 | 5.49 | **5.02** |
| 15 | 7.06 | $13.48^{+1}$ | 17.15 | $44.83^{+1}$ | 4.69 | 4.16 | 3.99 | 4.38 | 3.78 | **3.50** |
| 20 | 5.47 | 6.62 | $4.15^{+1}$ | 15.14 | 3.70 | 3.32 | 3.17 | 3.57 | 3.19 | **3.03** |
| 50 | 3.27 | 2.65 | $\mathbf{2.55}^{+1}$ | $2.57^{+2}$ | 2.70 | 2.62 | 2.59 | 2.70 | 2.61 | 2.59 |
| | | | FID for various DEIS with $\kappa = 3$ in Eq. (42) | | | | | | | |
| 5 | 32.11 | $364^{+1}$ | $393^{+1}$ | $383^{+3}$ | 28.87 | 28.58 | 28.62 | 25.78 | 23.66 | **23.38** |
| 10 | 13.18 | 135 | $199^{+2}$ | $298^{+2}$ | 9.89 | 9.38 | 9.33 | 7.74 | 6.20 | **5.77** |
| 15 | 7.92 | $42.04^{+1}$ | 99.64 | $122^{+1}$ | 5.41 | 4.99 | 4.91 | 4.48 | 3.65 | **3.37** |
| 20 | 5.92 | 17.05 | $16.66^{+1}$ | 64.40 | 4.04 | 3.69 | 3.60 | 3.54 | 3.05 | **2.86** |
| 50 | 3.36 | 2.77 | $2.57^{+1}$ | $2.71^{+2}$ | 2.73 | 2.63 | 2.60 | 2.67 | 2.59 | **2.57** |
| | | | FID for various DEIS with Eq. (44) | | | | | | | |
| 5 | 54.85 | $230^{+1}$ | $382^{+1}$ | $370^{+3}$ | 51.94 | 51.62 | 51.58 | 43.84 | 39.91 | **38.76** |
| 10 | 19.80 | 23.35 | $25.08^{+2}$ | $82.17^{+2}$ | 14.63 | 13.43 | 13.07 | 11.14 | 7.78 | **6.02** |
| 15 | 11.29 | $5.63^{+1}$ | 7.46 | $8.90^{+1}$ | 7.31 | 6.28 | 5.90 | 5.89 | 4.35 | **3.71** |
| 20 | 7.91 | 3.84 | $3.05^{+1}$ | 4.14 | 4.91 | 4.19 | 3.91 | 4.23 | 3.35 | **3.00** |
| 50 | 3.82 | 2.60 | $\mathbf{2.56}^{+1}$ | $2.59^{+2}$ | 2.86 | 2.70 | 2.64 | 2.79 | 2.63 | 2.58 |

Table 7: DEIS for VPSDE on CIFAR10 with $t_0 = 10^{-4}$ in Eq. (42)

| NFE | DDIM | $\rho$2Heun | $\rho$3Kutta | $\rho$4RK | $\rho$AB1 | $\rho$AB2 | $\rho$AB3 | $t$AB1 | $t$AB2 | $t$AB3 |
|-----|------|-------------|--------------|-----------|-----------|-----------|-----------|--------|--------|--------|
| | | | FID for various DEIS with $\kappa = 7$ in Eq. (43) | | | | | | | |
| 5 | 53.20 | $108^{+1}$ | $185^{+1}$ | $193^{+3}$ | 47.56 | 46.36 | 46.13 | 36.98 | 28.76 | **25.76** |
| 10 | 18.99 | 18.75 | $20.27^{+2}$ | $54.92^{+2}$ | 13.38 | 11.84 | 11.21 | 10.92 | 8.26 | **6.87** |
| 15 | 10.91 | $4.89^{+1}$ | 6.31 | $9.79^{+1}$ | 6.90 | 5.86 | 5.42 | 6.12 | 4.86 | **4.33** |
| 20 | 7.81 | 3.50 | $\mathbf{2.97}^{+1}$ | 3.92 | 4.84 | 4.10 | 3.80 | 4.48 | 3.69 | 3.38 |
| 50 | 3.84 | 2.60 | $\mathbf{2.58}^{+1}$ | $2.59^{+2}$ | 2.86 | 2.69 | 2.64 | 2.82 | 2.66 | 2.61 |

Table 8: DEIS for VPSDE on CIFAR10 with $t_0$ and time scheduling suggested by Karras et al. (2022)

### H.9   MORE REUSLTS ON VESDE

Though VESDE does not achieve the same accelerations as VPSDE, our method can significantly accelerate VESDE sampling compared with previous method for VESDE. We show the accelerated FID for VESDE on CIFAR10 in Tab. 15 and sampled images in Fig. 10.

### H.10   CHECKPOINT USED AND CODE LICENSES

Our code will be released in the future. We implemented our approach in Jax and PyTorch. We have also used code from a number of sources in Tab. 16.

| Method | FID with various NFE | | | | | | | | |
|---|---|---|---|---|---|---|---|---|---|
| | 5 | 10 | 20 | 30 | 50 | 100 | 200 | 500 | 1000 |
| Euler | 246.16 | 90.52 | 27.38 | 14.99 | 8.46 | 4.96 | 3.54 | 2.81 | 2.62 |
| + EI | 283.67 | 216.47 | 137.20 | 100.74 | 68.03 | 37.93 | 18.81 | 6.66 | 3.69 |
| $+\epsilon_\theta$ | 42.38 | 17.23 | 9.71 | 7.56 | 5.76 | 4.24 | 3.37 | 2.83 | 2.67 |
| + Poly | 30.67 | 10.41 | 6.50 | 5.13 | 4.06 | 3.07 | 2.69 | 2.58 | 2.57 |
| + Opt $\{t_i\}$ | 15.37 | 5.02 | 3.03 | 2.70 | 2.59 | 2.57 | 2.56 | 2.56 | 2.56 |

Table 9: Quantitative comparison in Fig. 5 for introduced ingredients, Exponential Integrator (EI), $\epsilon_\theta$-based score parameterization, polynomial extrapolation, and optimizing time discretization $\{t_i\}$, where we change uniform stepsize to quadratic one $t_0 = 10^{-4}$. We include Tabs. 6 to 8 for more ablation studies regarding time discretization.

| Method | FID with various NFE | | | | | | | | |
|---|---|---|---|---|---|---|---|---|---|
| | 5 | 10 | 20 | 30 | 50 | 100 | 200 | 500 | 1000 |
| Uniform | 246.16 | 90.52 | 27.38 | 14.99 | 8.46 | 4.96 | 3.54 | 2.81 | 2.62 |
| Quadratic | 294.01 | 138.73 | 39.82 | 19.26 | 8.49 | 3.96 | 2.88 | 2.61 | 2.57 |

Table 10: Effects of different timesteps on the Euler method. We use $t_0 = 10^{-4}$ which has lower FID score compared with the default $t_0 = 10^{-3}$ (Song et al., 2020b) in the experiments.

We list the used checkpoints and the corresponding experiments in Tab. 17.

# I    MORE RESULTS FOR IMAGE GENERATION

| NFE | 14 | 26 | 32 | 38 | 50 | 62 | 88.2 | 344 |
|-----|-----|-----|-----|-----|-----|-----|-----|-----|
| FID | 61.11 | 36.64 | 15.18 | 9.88 | 6.32 | 2.63 | 2.56 | 2.55 |

Table 11: Quantitative performance of RK45 ODE solver with $t_0 = 10^{-4}$ in Fig. 5.

| FID\NFE


Method | 5 | 10 | 20 | 50 |
|------|------|------|------|------|
| A-DDIM | 51.47 | 14.06 | 6.74 | 4.04 |
| 1-iPNDM | 30.13 | 13.01 | 8.25 | 5.65 |
| 2-iPNDM | 84.00 | 10.45 | 6.79 | 4.73 |
| 3-iPNDM | 105.38 | 14.03 | 5.79 | 4.24 |
| $t$AB1-DEIS | 20.45 | 8.11 | 4.91 | 3.88 |
| $t$AB2-DEIS | 18.87 | 7.47 | 4.66 | 3.79 |
| $t$AB3-DEIS | **18.43** | **7.12** | **4.53** | **3.78** |

Table 12: Comparison with A-DDIM on the checkpoint and time scheduling provided by (Bao et al., 2022) on CIFAR10

| FID\NFE


Method | 5 | 10 | 20 | 50 |
|------|------|------|------|------|
| iPNDM | 54.62 | 15.32 | 9.26 | 8.26 |
| DDIM | 49.08 | 23.52 | 13.69 | 9.44 |
| $t$AB1-DEIS | 34.69 | 13.94 | 9.55 | 8.41 |
| $t$AB2-DEIS | 29.50 | 11.36 | 8.79 | 8.29 |
| $t$AB3-DEIS | **28.09** | **10.55** | **8.58** | **8.25** |

Table 13: Sampling quality on VPSDE ImageNet32 $\times$ 32 with the checkpoint provided by Song et al. (2021a). Blackbox ODE solver reports FID 8.34 with ODE tolerance $1 \times 10^{-5}$ (NFE around 130).

| Dataset | FID\NFE


Method | 5 | 10 | 20 | 50 |
|------|------|------|------|------|------|
| | PNDM | - | - | 7.60±0.12 | 3.51±0.03 |
| | iPNDM | 59.87±1.01 | 7.78±0.18 | 5.58±0.11 | 3.34±0.04 |
| CELEBA | DDIM | 30.42±0.87 | 13.53±0.48 | 6.89±0.11 | 4.17±0.04 |
| | $t$AB1-DEIS | 26.65±0.63 | 8.81±0.23 | 4.33±0.07 | 3.19±0.03 |
| | $t$AB2-DEIS | 25.13±0.56 | 7.20±0.21 | 3.61±0.05 | 3.04±0.02 |
| | $t$AB3-DEIS | **25.07±0.49** | **6.95±0.09** | **3.41±0.04** | **2.95±0.03** |

Table 14: Mean and standard deviation of multiple runs with 4 different random seeds on the checkpoint and time scheduling provided by Liu et al. (2022) on CELEBA.

| SDE | FID\NFE


Method | 5 | 10 | 20 | 50 |
|------|------|------|------|------|------|
| | $t$AB0-DEIS | 103.52±2.09 | 46.90±0.38 | 27.64±0.05 | 19.86±0.03 |
| | $t$AB1-DEIS | **56.33±0.87** | 26.16±0.12 | 18.52±0.03 | 16.64±0.01 |
| VESDE | $t$AB2-DEIS | 58.65±0.25 | **20.89±0.09** | 16.94±0.03 | 16.33±0.02 |
| | $t$AB3-DEIS | 96.70±0.90 | 25.01±0.03 | **16.59±0.03** | **16.31±0.02** |

Table 15: FID results of DEIS on VESDE CIFAR10. We note the Predictor-Corrector algorithm proposed in (Song et al., 2020b) have $\geq 100$ FID if sampling with limited NFE budget ($\leq 50$).

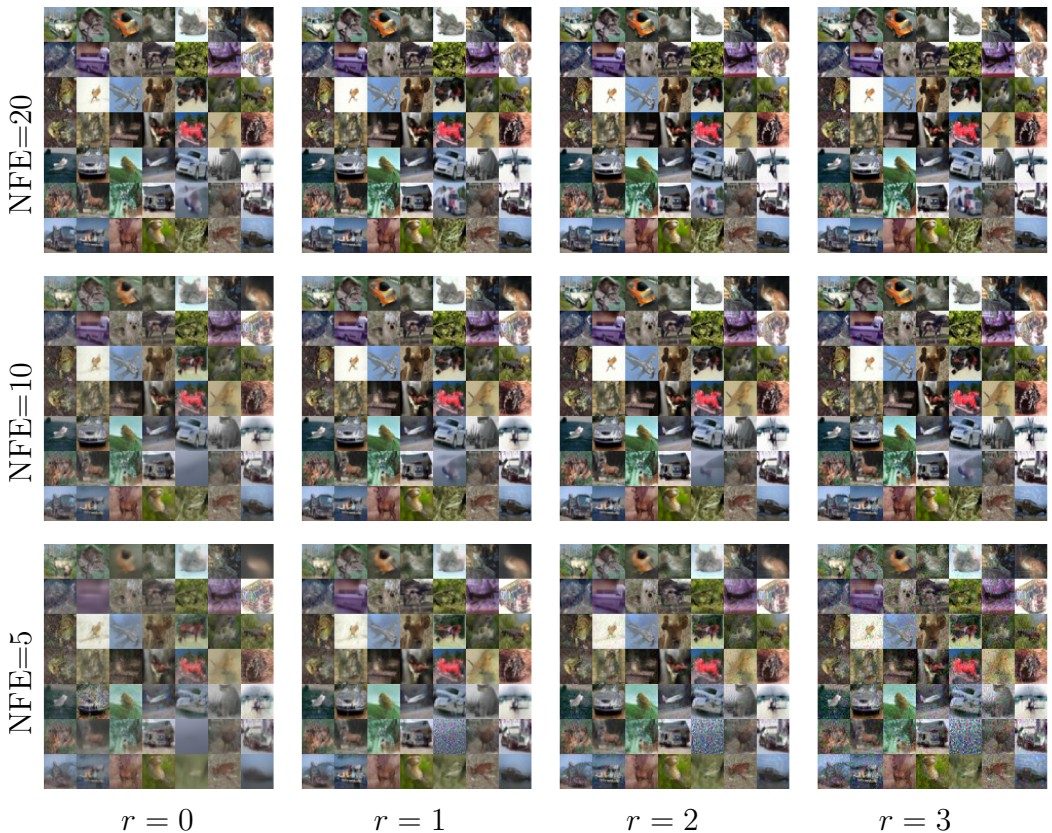

Figure 10: Generated images with $t$AB$r$-DEIS on VESDE CIFAR10.

| URL | License |
|---|---|
| https://github.com/yang-song/score_sde | Apache License 2.0 |
| https://github.com/luping-liu/PNDM | Apache License 2.0 |
| https://github.com/CompVis/latent-diffusion | MIT |
| https://github.com/baofff/Analytic-DPM | Unknown |

Table 16: Code Used and License

| Experiment | Citation | License |
|---|---|---|
| CIFAR10 Tabs. 2, 6 to 8, 11, 14 and 15, FFHQ Fig. 1 | (Song et al., 2020b) | Apache License 2.0 |
| CIFAR10 Tab. 12 | (Bao et al., 2022) | Unknown |
| CELEBA Tab. 14 | (Liu et al., 2022) | Apache License 2.0 |
| ImageNet $32 \times 32$ Tab. 13 | (Song et al., 2021a) | Unknown |
| Text-to-image | (Rombach et al., 2021) | MIT |

Table 17: Checkpoints for experiments

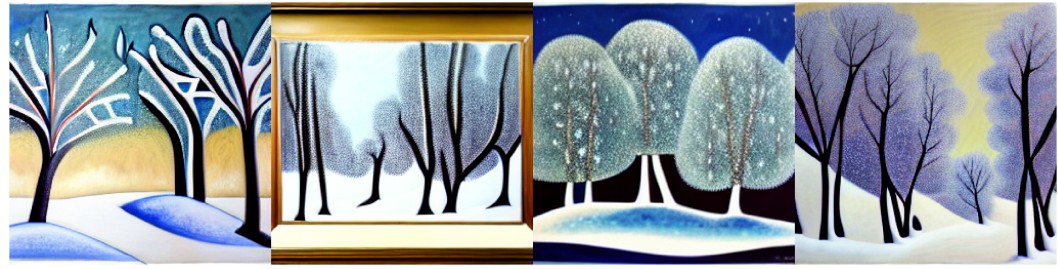

Figure 11: Generated images with text "A artistic painting of snow trees by Pablo Picaso, oil on canvas" (15 NFE)

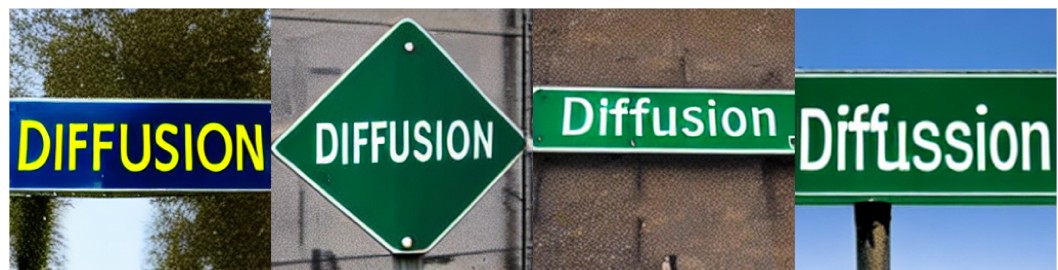

Figure 12: Generated images with text "A street sign that reads Diffusion" (15 NFE)

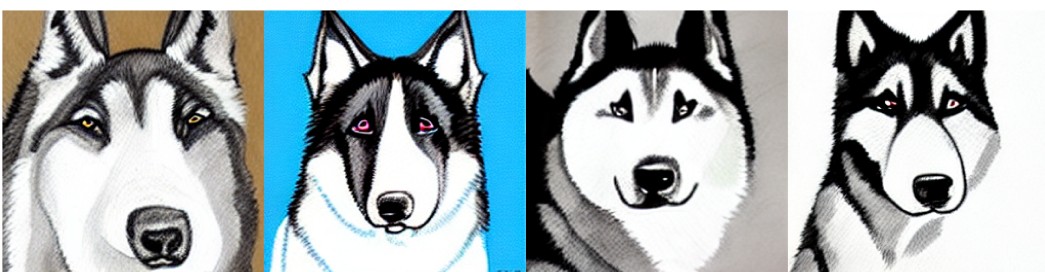

Figure 13: Generated images with text "The drawing of a funny husky" (15 NFE)

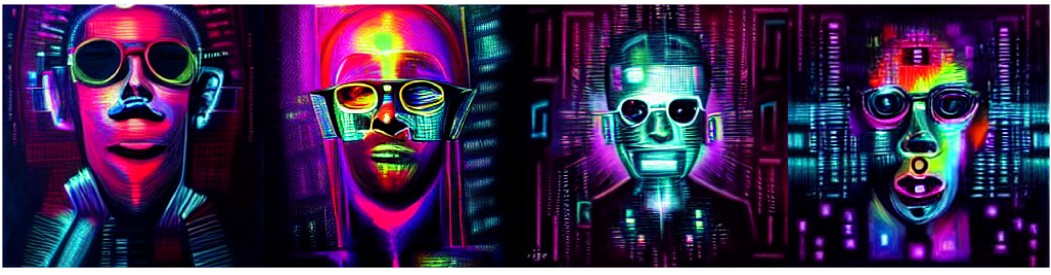

Figure 14: Generated images with text "Cyber punk oil painting" (15 NFE)

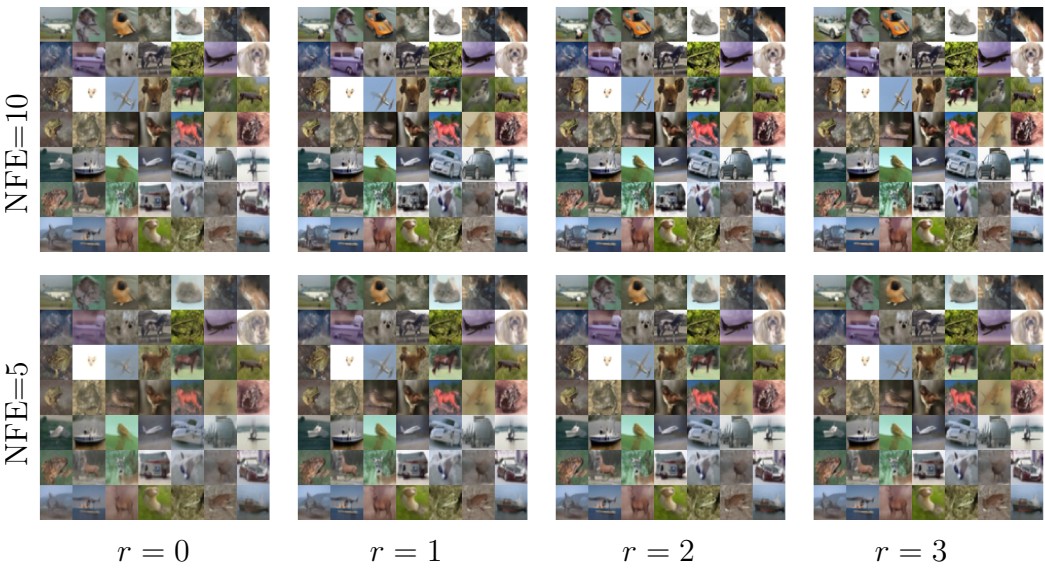

Figure 15: Generated images with DEIS on VPSDE CIFAR10.

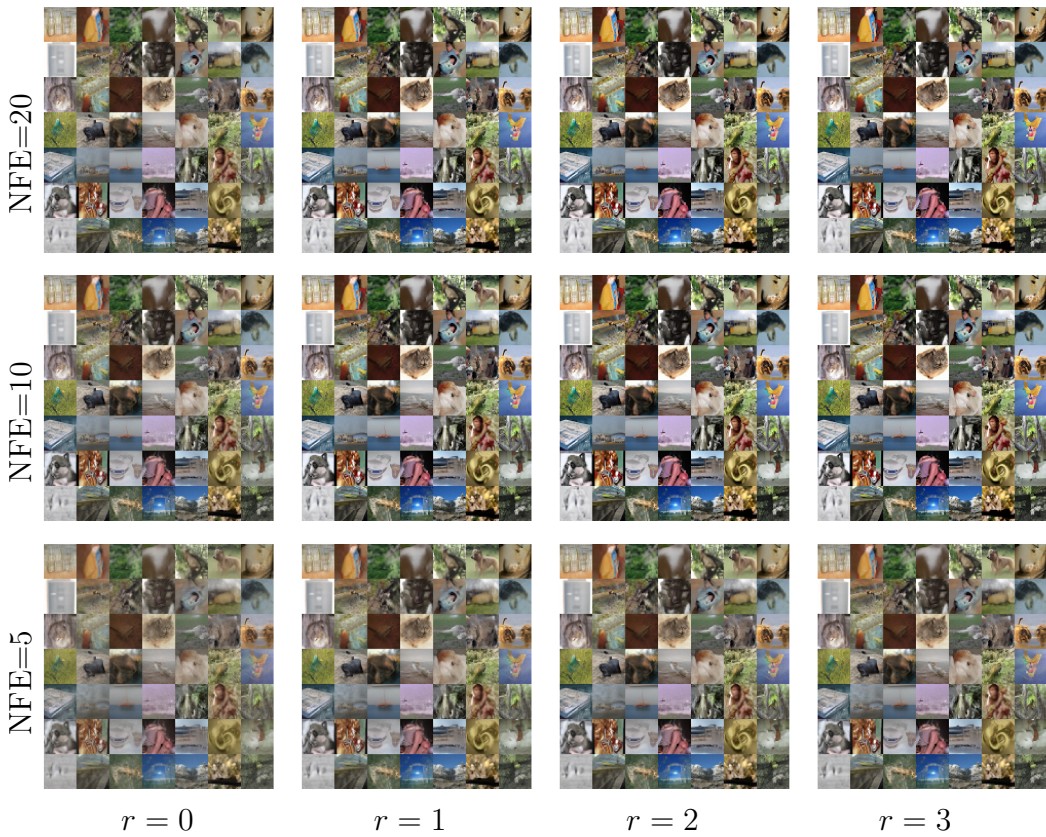

Figure 16: Generated images with DEIS on VPSDE ImageNet $32 \times 32$.

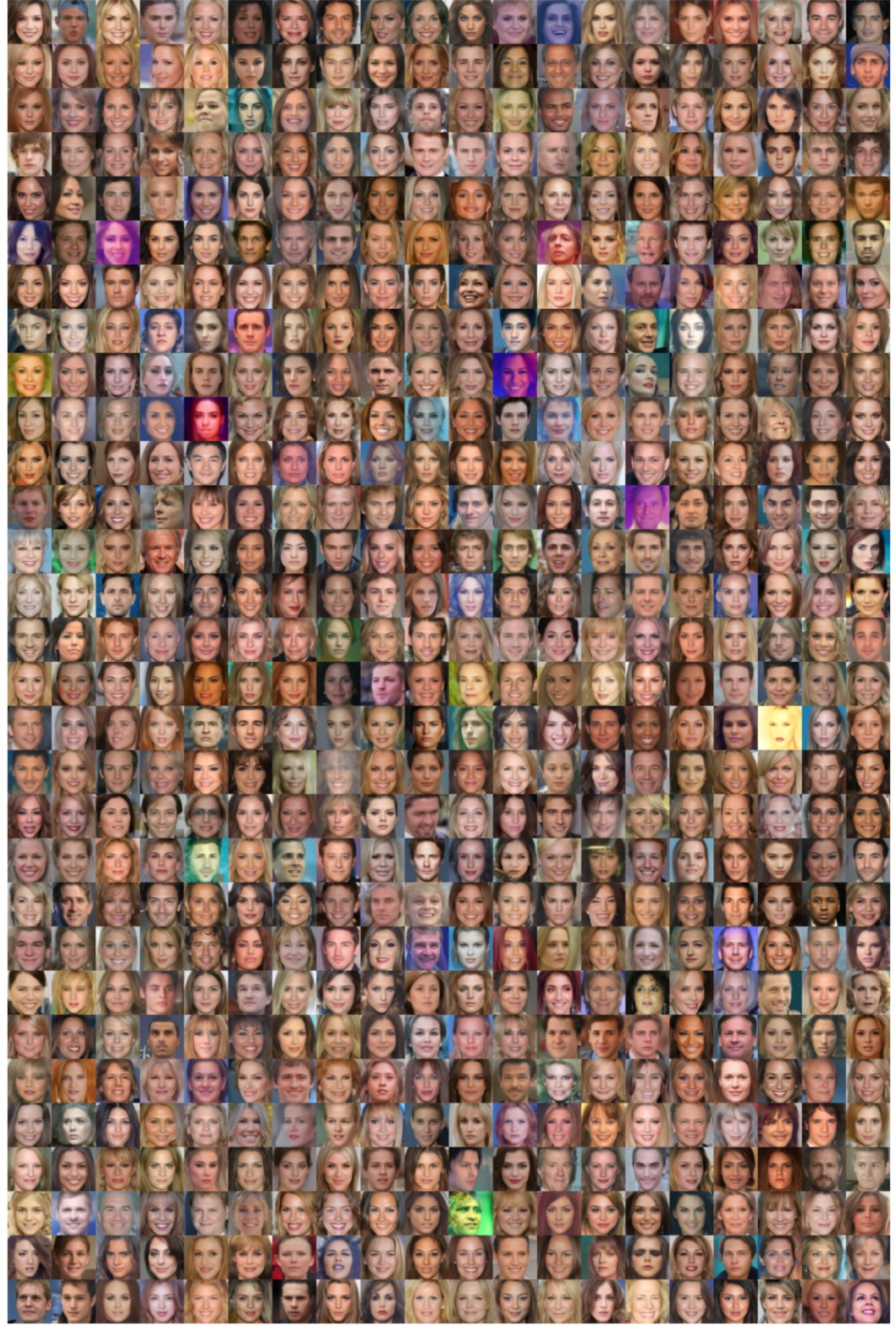

Figure 17: Generated images with DEIS on VPSDE CelebA (NFE 5).

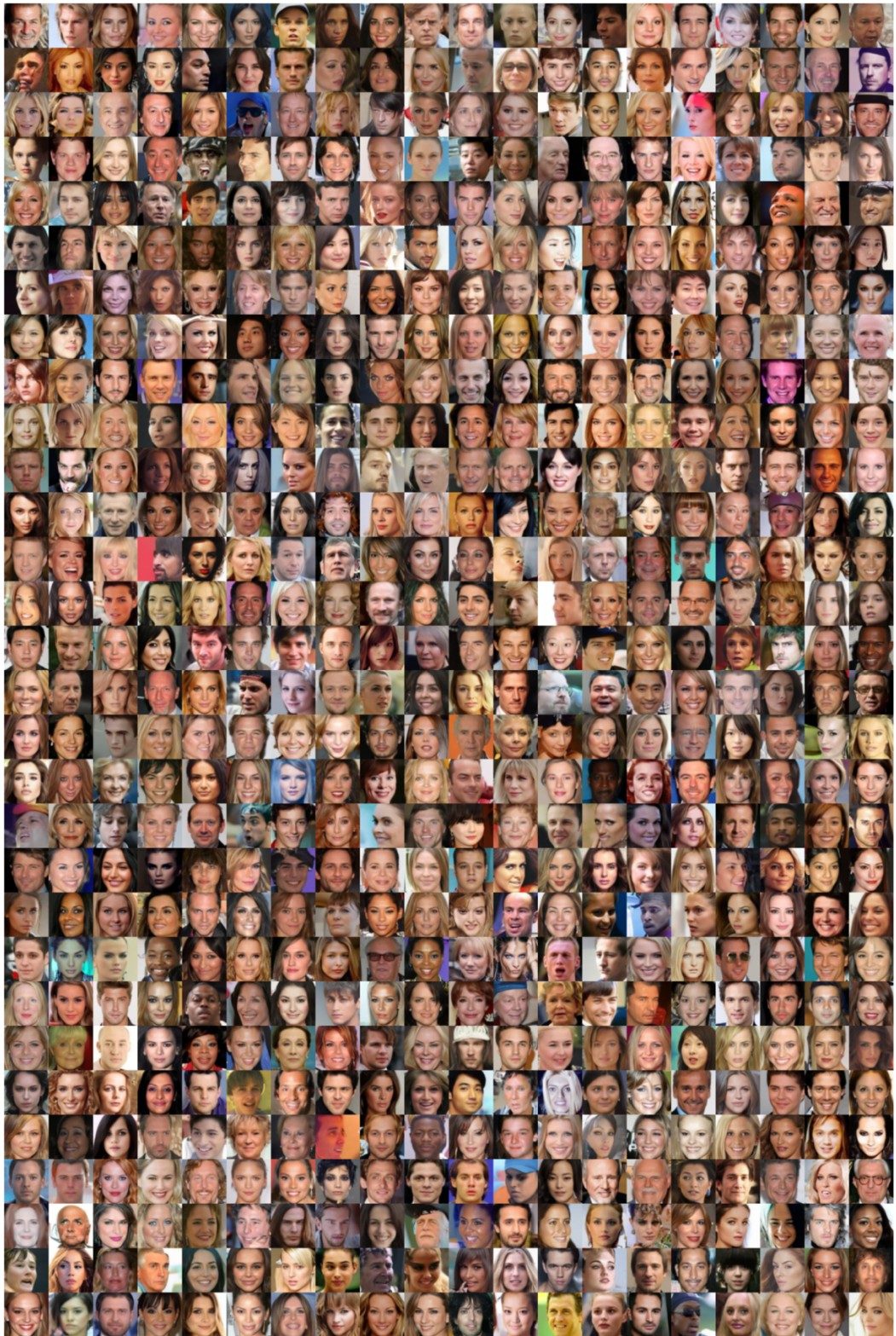

Figure 18: Generated images with DEIS on VPSDE CelebA (NFE 10).

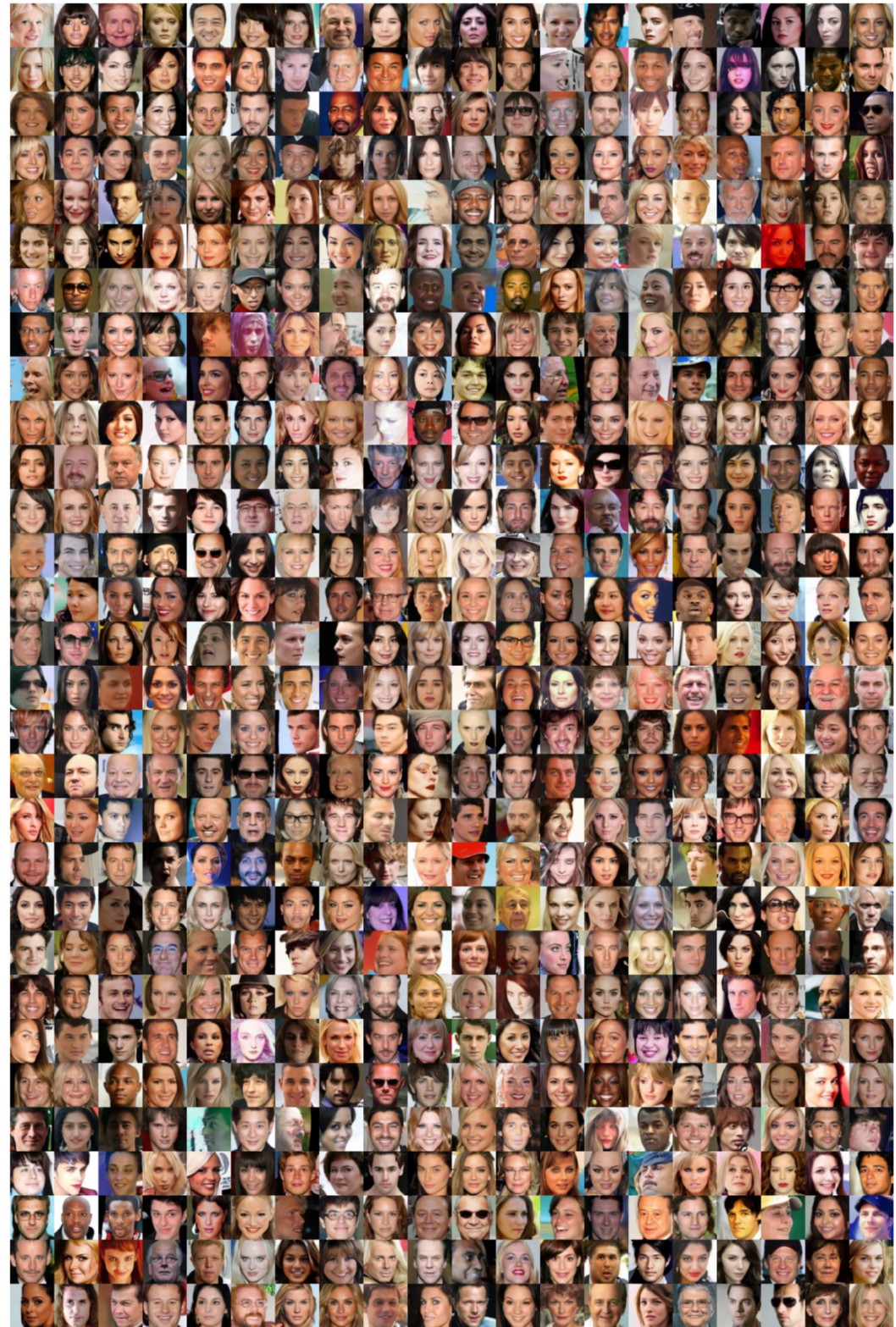

Figure 19: Generated images with DEIS on VPSDE CelebA (NFE 20).

