# OpenReview forum: "Fast Sampling of Diffusion Models with Exponential Integrator"
_ICLR.cc/2023/Conference — ICLR 2023 poster_

### Official Review · Reviewer_HoiQ · 2022-10-23

**Confidence:** 3
**Correctness:** 3
**Technical Novelty And Significance:** 3
**Empirical Novelty And Significance:** 2
**Recommendation:** 6

**Clarity, Quality, Novelty And Reproducibility:**

The work is well presented. The quality is good in terms of analysis and writing. The paper is clean and detailed enough for reproducibility. However, there are a few of concurrent works such as DPM-Solver and PLMS sampler are not compared in the current version.

**Strength And Weaknesses:**

Strengths:
1. The paper is well written. The flow is easy to follow and the contributions of the paper is clearly presented.
2. The background knowledge on diffusion model is well presented and the readers can easily get the main claims of the paper, which is the sampling efficiency.
3. I like the analysis conducted by the authors. The claims are supported by the experiments results.

Weakness:
1. The paper only conducted experiment on easy & object centric datasets: CIFAR and ImageNet with 32x32 size. However, the main advantages of DM are on complex scenes where the generated images contain multiple objects. Besides, one of the well known disadvantages of DM is the high resolution images. Could the proposed algorithm achieve promising results on applied to the above mentioned cases?

2. In Appendix B, authors claim that DPM-Solver is a concurrent work. The authors claim that DPM-solver is a special case of DEIS. So, how about the comparison with DPM-Solver? Does the more general exploration of DEIS outperform DPM-Solver?

3. Recent popular LDM models use PLMS solver and achieves even faster sampling speed. How the proposed DEIS compared to PLMS sampler?

**Summary Of The Paper:**

The paper targets at improving the sampling efficiency of diffusion model via a semi-linear ODE methods, termed as Diffusion Exponential Integrator Sampler (DEIS). The paper is well motivated with experiment observations that the changes of noise predicted by the neural network is small for most of the sampling steps as shown in Fig. 4a in the paper. Thus the values can be reused. The author also explores a variety of functions by measuring the  discretization error. The author finally proposes a principled guideline on achieving fast sampling of DM models with High order polynomial and . The experiments on CIFAR-10 and ImageNet with image size downsampled to 32x32 support the main claims of the paper.

**Summary Of The Review:**

Overall, I like the paper's style and I think the authors have put in enough efforts on this work. However, the advantages of the proposed methods are not clean enough. The experiments are relatively weak in terms of both dataset and selected baselines. Please refer to the `strength & weakness' section for the details. Nevertheless, the paper have put in enough analysis that reveals some of the properties of DM. I would vote for board-line accept for this paper.

---

> ### Author Response · Authors · 2022-11-17
> **Response**
>
> ********Q: Recent popular LDM models use PLMS solver and achieve even faster sampling speed. How the proposed DEIS compared to PLMS sampler?********
>
> A: We believe the suggested PLMS is PNDM [1]. In fact, for the popular LDM models, several researchers have integrated iPNDM~(improved PNDM/PLMS/LMS) **proposed in our work** into popular repositories, such as [crowsonkb/v-diffusion-pytorch](https://github.com/crowsonkb/v-diffusion-pytorch/blob/987f8985e38208345c1959b0ea767a625831cc9b/README.md?plain=1#L135), [huggingface/diffusers](https://github.com/huggingface/diffusers/blob/main/src/diffusers/schedulers/scheduling_lms_discrete.py#L138).
>
> We have included a comparison between PNDM and  DEIS in CELEBA and CIFAR10. In all settings, DEIS is better than iPNDM and iPNDM is better than PNDM.
>
> We are happy that iPNDM(iPLMS) is popular in the open-source community and we sincerely hope our work and contributions can be valued.
>
> ********Q: Could the proposed algorithm achieve promising results on applied to the large resolution datasets?********
>
> A: DEIS can be applied to any diffusion models, regardless of the resolutions. We have included more experiments on CelebA 64x64, ImagNet 64x64 and LSUN Bedroom 256x256 in the paper, where we demonstrate DEIS can achieve promising results.
>
> ********Q: Authors claim that DPM-Solver is a concurrent work. The authors claim that DPM-solver is a special case of DEIS. So, how about the comparison with DPM-Solver? Does the more general exploration of DEIS outperform DPM-Solver?********
>
> A:
>
> 1. We thank the reviewer’s comments, the discussions in openreview, and comments from DPM-Solver author. Though the two algorithms share a lot of similarities, we find they have different update schemes. We have made corresponding corrections in our paper and included more comparisons with DPM-Solver quantitatively and qualitatively. We find $t$AB-DEIS outperforms DPM-Solver2 and DPM-Solver 3 in general.
> 2. About **originality**. In fact, DEIS appears on arXiv around 2 months easier than DPM-Solver~(first available on Jun 2nd). Therefore, we claim these are concurrent work if DEIS is not a “existing” methods for DPM-Solver.
>
> ********Q: The experiments are relatively weak in terms of both dataset and selected baselines.********
>
> A: Regarding datasets, we have added ImageNet 64x64 and Bedroom 256x256 for comparison. We respectfully disagree with the point regarding selected baselines. The popular PNDM~(PLMS) and Heun method [2] are included in our empirical comparison (see above discussion).
>
> [1] L. Liu. Pseudo Numerical Methods for Diffusion Models on Manifolds
>
> [2] T. Karras et al. Elucidating the Design Space of Diffusion-Based Generative Models

---

### Official Review · Reviewer_xTdk · 2022-10-24

**Confidence:** 4
**Correctness:** 3
**Technical Novelty And Significance:** 3
**Empirical Novelty And Significance:** 3
**Recommendation:** 8

**Clarity, Quality, Novelty And Reproducibility:**

**Clarity**
I find the use of matrix notation throughout makes the paper more difficult to understand. Moreover, the matrix notation is never actually needed, since in Table 1, all of these coefficients are multiples of the identity.

It's unclear to me why you need the general form in eq 4. with lambda nonzero. In Section 5, you mention discretizing this with lambda nonzero 'does not work well in practice', and 'We do not pursue the discretization of the SDE Eq. (4) further in this paper and leave it for future'. Why include it at all then? You also include motivation as to why we should only consider the lambda = 0 case ('From this perspective, the ODE with λ = 0 is the best option since it minimizes the weight of nonlinear term.'), but why not make this case earlier and just consider the simplified version of eq. 4?

Overall there is some difficulty parsing the paper. This includes some awkward phrasing (e.g. abstract: 'a fast sampling method for DMs with a much less number of steps' ->  'a fast sampling method for DMs with fewer steps') and some sentences which don't make sense (e.g. 4: 'Absorbing useful structures such as semi-linear structures through a transformation so that the transformed ODE does not possess any useful structure that one can utilize to improve generic ODE methods.'). It would be good to revise some of this.

The polynomial extrapolation and associated coefficient computation in Ingredient 3 (eqs 14 and 15) might benefit from an explicit algorithm description i.e. to make clear that the coefficients are computed once in advance when a particular discretization is chosen (if I understand correctly).

**Originality**
The paper has a lot in common with DPM-Solver https://arxiv.org/abs/2206.00927 (Lu et al), as you've noted in Appendix B. I appreciate that this is concurrent work, but I feel this comparison should be made in the main body of the paper, rather than in the appendix, especially since you claim these methods are special cases of your approach. It would be useful to tease out the exact correspondence to this paper, and explain the choices in https://arxiv.org/abs/2206.00364 (Karras et al) in terms of your formulation, as well as compare to both empirically.

**Strength And Weaknesses:**

**Strengths**
The formulation of the solver is relatively well-structured and motivated. The three ingredients proposed in section 3.2 and the related ablation of each on the toy problem is neat. The explanation of DDIM in terms of a first order method for the semi-linear ODE is nice, although a similar argument was already made in DPM-Solver (Lu et al) (although I acknowledge it was concurrent work, more discussion below). The reparameterization in section 4 removing the linear term in section 4 is also neat, but again the discussion overlaps with Karras et al (also discussed more below). The experimental results, primarily Table 2, are convincing, especially showing the worth of the higher order methods.

**Weaknesses**
The experiments section is quite concise. There is quite a lot of additional content in appendices G and H. It would be good to compare to more methods in the main paper (as well as the concurrent work discussed below) and not relegate this to the appendix. It would also be good to expand on the use of the proposed sampler for pretrained models like Stable Diffusion beyond qualitative samples -- showing improved performance here over competitive approaches would be a strong result.

Related to iPNDM: 'We further propose Improved PNDM (iPNDM) that uses lower order multistep methods for the first few steps in sampling to avoid expensive warming start.' Isn't it the case that for linear multistep methods the expensive warm start is necessary? i.e. if you don't warm start an order K methods with steps of at least order (K - 1), the resulting method is bottlenecked by the lowest order used for the warm start?

**Nits**
- Intro: 'DM is not hyperparameter sensitive' - maybe less sensitive than GANs, otherwise this is too strong a statement.
- Intro: 'extremely slow sampling' - again, 'extreme' is probably too strong a statement, compared with e.g. AR sampling
- Figure 2: the vertical-axis isn't labeled
- 7: 'DEIS also works amazingly in practice' - again too strong

**Summary Of The Paper:**

This paper proposes a sampler for diffusion models based on the exponential integrator and the semi-linear structure of the probability flow ODE. The authors demonstrate that this sampler maintains strong performance for low numbers of function evaluations.

**Summary Of The Review:**

Overall I think this is a well-motivated paper with convincing empirical results. A more thorough discussion and comparison of competing methods, as well as some restructuring to move results from the appendix would improve the paper further.

Edit post-rebuttal: I'd like to thank the authors for their response. Some of the main criticism of the paper concerns concurrent work, which I feel the authors have attempted to address. Both the concurrent work and this paper constitute useful contributions, and I think this paper also deserves to appear in proceedings, even if the overall presentation is perhaps less clear. I've raised my score accordingly.

---

> ### Author Response · Authors · 2022-11-17
> **Response**
>
> ************Q: Originality, It would be useful to tease out the exact correspondence to this paper, and explain the choices in (Karras et al) in terms of your formulation, as well as compare to both empirically.************
>
> A:
>
> 1. ************Originality.************ In fact, DEIS appears on arXiv around **2 months easier** than DPM-Solver [1] (first available on Jun 2nd), Elucidating Diffusion Model [2] (first available on Jun 1st and acknowledged DEIS). Therefore, we claim [1][2] and DEIS are concurrent work if DEIS is not a “existing” method for [1][2].
> 2. For DPM-Solver,  we have made modifications in our draft to correct our claim. The DPM-Solver is slightly different from $\rho$RK-DEIS, theoretically and empirically. We have added a comprehensive comparison between DEIS and the DPM-Solver. We find $t$AB-DEIS outperforms DPM-Solver and generates better samples based on our experiments.
> 3. For Elucidating Diffusion Model, the proposed second Heun method is exactly $\rho$2Heun-DEIS in our work. The connection and exact correspondence are explained in our discussion (Appendix B). We **have also included** it in our empirical comparison. We find 3order Kutta methods outperform 2Heun when we have a relatively large NFE budget, and $t$AB-DEIS outperforms $\rho$RK-DEIS.
> 4. Please check the discussion section (Appendix B) in our draft.
>
> **Q: I find the use of matrix notation throughout makes the paper more difficult to understand. Moreover, the matrix notation is never actually needed, since in Table 1, all of these coefficients are multiples of the identity.**
>
> A: The DEIS method we develop is applicable to general diffusion models even though for most diffusion models the coefficients are multiples of the identity.
>
> Recently, a couple of works have investigated more diffusion models, such as critically-damped Langevin diffusion (CLD) **[3], and the Blurring diffusion models (BDM) [4] where matrix ODE is necessary.
>
> We use matrix ODE to keep the generality of our method. We have added a sentence following Eq (1) to clarify this point.
>
> ********Q: if you don't warm start order K methods with steps of at least order (K - 1), the resulting method is bottlenecked by the lowest order used for the warm start?********
>
> A: We thank the reviewer for pointing this out. As we discussed in the appendix, the theoretical results for iPNDM may be not be as elegant as standard multistep methods, and the accuracy could be bottlenecked by the lowest order warm start step. However, we will not pursue this theoretical question further as it is not our focus. We focus on a sampling scheme with large step sizes and **very small NFE**, where the convergence of any numerical scheme is **out of question**. Empirically, we find iPNDM has better sampling quality than PNDM.
>
> **********Q: It's unclear to me why you need the general form in eq 4. with lambda nonzero. In Section 5, you mention discretizing this with lambda nonzero 'does not work well in practice’**********
>
> A: The exponential integrator (EI) is still applicable to SDE and **achieves better performance** than all SDE sampler without EI, however, the performance is still not as good as that of DEIS for ODE. This observation is consistent with existing literatures on diffusion model sampling where ODE based methods normally have better performances [5]. This observation however does not exclude the possibility that SDE methods can eventually be better than ODE methods for diffusion model sampling. In fact, all the existing theoretical studies investigating the sampling complexity of diffusion models are all based on SDE [6].
>
> In this paper, we also establish some theoretical results on SDE based approaches and provide some explanations on why these methods could have worse performance when NFE is small. We believe these explanations and ``negative results'' are important steps to develop even better algorithms and are beneficial to the community.
>
> ********Q: The experiments section is quite concise. There is quite a lot of additional content in appendices G and H. It would be good to compare to more methods in the main paper (as well as the concurrent work discussed below) and not relegate this to the appendix.********
>
> A: We have reorganized the manuscript and added more experiments with pre-trained ImageNet models and LSUN Bedroom models in the main paper. We have moved the comparison on the CelebA dataset to the main paper, and it shows our methods are better than existing approaches.

---

> > ### Author Response · Authors · 2022-11-17
> > **Response2**
> >
> > **Q: The polynomial extrapolation and associated coefficient computation in Ingredient 3 (eqs 14 and 15) might benefit from an explicit algorithm description.**
> >
> > A: We have highlighted a sentence following (15) that clarifies the update scheme.
> >
> > [1] C. Lu et al. DPM-Solver: A Fast ODE Solver for Diffusion Probabilistic Model Sampling in Around 10 Steps
> >
> > [2] T. Karras et al. Elucidating the Design Space of Diffusion-Based Generative Models
> >
> > [3] T.Dockhorn Score-Based Generative Modeling with Critically-Damped Langevin Diffusion
> >
> > [4] E. Hoogeboom et al. Blurring Diffusion Models
> >
> > [5] J. Song et al. Denoising Diffusion Implicit Models
> >
> > [6] H. Lee et al. Convergence of Score-based Generative Modeling for General Data Distributions

---

### Official Review · Reviewer_xzme · 2022-10-24

**Confidence:** 3
**Correctness:** 3
**Technical Novelty And Significance:** 3
**Empirical Novelty And Significance:** 3
**Recommendation:** 6

**Clarity, Quality, Novelty And Reproducibility:**

I liked the idea of using EI and polynomial extrapolation for the fast sampling in the ODE version of diffusion models, which is novel and original to me. However, I think the presentations of the idea and experiment results should be largely improved. Lastly, I think the work provided sufficient details for reproducing the main results.

**Strength And Weaknesses:**

Strengths:
- The proposed idea from the exponential integrator perspective is interesting and novel to me. The motivation of reducing the discretization error using polynomial extrapolation is sound and intuitive.
- This work provides a good insight into the understanding of DDIM from the numerical discretization perspective. That is, applying EI into the noise prediction reparameterization in VPSDE without the polynomial extrapolation becomes the DDIM sampling.
- Experimental results on CIFAR-10 show the effectiveness of the proposed method: the best variant of DEIS largely outperforms previous ODE-based training-free methods (DDIM, A-DDIM, PNDM) when the NFE is less than 20.

Weaknesses:
- The presentation of the proposed method is not clear. I understand that this work tries to make the main idea well-motivated and general, but I think many design choices that do not work well can be moved to the appendix, such that we understand the core idea more easily. Below are some details: 1) if EI works better than Euler with the noise prediction reparameterization (while it works worse than Euler with score prediction reparameterization), we can present the idea with the noise prediction reparameterization, which has been commonly used in most diffusion models for images. The analysis for different network reparameterizations can be deferred to the appendix. 2) I don’t fully understand why we need Proposition 3. In other words, why do we need to transform Eq. (10) into “a simple non-stiff ODE”, if the resulting $\rho$AB-DEIS is no better than $t$AB-DEIS (Table 2)? Also, I don’t fully understand why it is a good thing to apply generic ODE solvers without “worrying about the semi-linear structure”. 3) If DEIS does not work well for SDE, we can defer most details in Section 5 to the appendix. 4) I’m not exactly sure what “+optimizing timesteps” in Figure 5 means. It seems that the main text doesn’t directly explain its meaning.
- The experiment settings and results are also confusing. Below are some details: 1) It mentions that the proposed method compares with other training-free baselines (DDIM, A-DDIM, PNDM) on several datasets. However, in the main text, we don’t see the results of A-DDIM and PNDM at all, and there is no quantitative result on other datasets except CIFAR-10. I suggest presenting these main experimental results in a more compact and coherent way instead of deferring most of them to the appendix. 2) In the experimental setting, I think several more recent training-free baselines (e.g. [1,2]) are missing. I noticed that the authors discussed their relationship with the proposed methods in the appendix. I would prefer to discuss these relevant baselines and show quantitative comparison results in the main text. 3) I appreciate that the authors considered ImageNet as one of the datasets to evaluate the proposed method. But I think ImageNet 32x32 is a bit low resolution and difficult to visually see the generation quality. I suggest similar to many previous works (e.g., [2,3]) that consider ImageNet with a resolution of at least 64x64 for obtaining both quantitative and qualitative results.
- Minor things: In section 2, when describing the conditional marginal distribution $p_{0t}(x_t|x_0)$, what does $\mu_t$ mean? I suggest replacing the term “Diffusion model” with “diffusion model”.


[1] Elucidating the Design Space of Diffusion-Based Generative Models, NeurIPS 2022.
[2] DPM-Solver: A Fast ODE Solver for Diffusion Probabilistic Model Sampling in Around 10 Steps, NeurIPS 2022.
[3] Progressive Distillation for Fast Sampling of Diffusion Models, ICLR 2022.


**Summary Of The Paper:**

This work proposed a fast sampling method, termed as diffusion exponential integrator (DEIS), for diffusion models with a new discretization of the reverse process. In particular, this work first investigates the existing ODE solvers for diffusion models and found that reducing the discretization error is crucial for fast sampling. This work then proposes a new discretization that is mainly based on the exponential integrator (EI) and polynomial extrapolation of the noise prediction function. Besides, this work shows that DDIM is a special case of DEIS. Experiments show that the proposed method can achieve 4.17 FID with 10 function evaluations on CIFAR-10, outperforming other ODE discretization based methods.


**Summary Of The Review:**

Overall, I think the idea is interesting and useful, but the idea and experimental results are poorly presented, which makes me not sure how it compares with previous relevant methods. My initial suggestion is “leaning to reject” but I’m open to adjusting my rating.

---

> ### Author Response · Authors · 2022-11-17
> **Response**
>
> ********Q: Writing. Many design choices that do not work well can be moved to the appendix, such that we understand the core idea more easily.  For example, If DEIS does not work well for SDE, …. The experiment settings and results are also confusing. Below are some details: 1) It mentions that the proposed method compares with other training-free baselines (DDIM, A-DDIM, PNDM) on several datasets. However, in the main text, we don’t see the results of A-DDIM and PNDM at all, and there is no quantitative result on other datasets except CIFAR-10.********
>
> A: We partially agree with the reviewer’s comments. We have moved part of the discussion on SDE (previously Section 5) into the appendix and have included more experiment results in the main paper.
>
> The exponential integrator (EI) is still applicable to SDE and **achieves better performance** than all the existing methods based on SDE without EI, however, the performance is still not as good as that of DEIS for ODE. This observation is consistent with existing literatures on diffusion model sampling where ODE based methods normally have better performances [1]. This observation however does not exclude the possibility that SDE methods can eventually be better than ODE methods for diffusion model sampling. In fact, all the existing theoretical studies investigating the sampling complexity of diffusion models are all based on SDE [2].
>
> In this paper, we also establish some theoretical results on SDE based approaches and provide some explanations on why these methods could have worse performance when NFE is small. We believe these explanations and ``negative results'' are important steps to develop even better algorithms and are beneficial to the community.
>
> ********Q: I don’t fully understand why we need Proposition 3. I don’t fully understand why it is a good thing to apply generic ODE solvers without “worrying about the semi-linear structure”.********
>
> A: We want to emphasize that **Proposition 3 is a key contribution** in this work.
>
> 1. **Theoretically**, compared with existing semi-linear ODE formulation $\frac{dx}{dt} = F_t x - \frac{1}{2}G_t G^T_t s_\theta(x, t)$, the transformed ODE in Proposition 3 $\frac{dy}{d\rho} = \epsilon_{\theta}(y, \rho)$ has no linear term and no time-varying coefficients on the RHS. The transformed ODE is known to be easier to solve numerically compared with semi-linear ODE. A typical update scheme for $x$ from $t$ to $t-\Delta t$ is accompanied by discretization errors of approximating time-varying $F_{\tau}$ by $F_t$ and  $G_{\tau}$ by $G_t$ for $\tau \in [t- \Delta t, t] in semi-linear ODE, while the ODE  in Proposition 3 does not have this issue as it handles those term analytically in Proposition 3. With Proposition 3, we can apply any out-of-shelf ODE solver and the **algorithm design becomes more flexible**. Also, it is easy to **accelerate the evaluation of negative likelihood** using high order solver on the transformed ODE in Proposition 3. In contrast, it is not clear how to use t-AB-DEIS on the original semi-linear ODE for this purpose.
> 2. ********Empirically performance.********
>     1. For diffusion models, directly applying the Euler method to the transformed ODE is exactly DDIM, which is much more efficient compared with applying Euler or even higher RK45 on semi-linear ODE. The comparison is also included in Figure 5 where the blue curve is Euler on semi-linear ODE while the green curve is DDIM.
>     2. With Proposition 3, we can directly use out-of-shelf ODE solvers without accounting for the semi-linear structure in algorithm design. Though motivated from different perspectives, applying second-order Huen reproduces Heun method introduced in [3]. Proposition 3 is much easier to integrate various higher-order methods. Moreover, we empirically find the 3 order method Kutta leads better samples compared with Heun when NFE is relatively large.
>     3. Numerically, methods present in Sec 3 $t$AB-DEIS relies on high-resolution integrator~(or analytic form for low order) in Eq-(14,15) to handle time-varying $F_t, G_t$ while $\rho$DEIS in Sec 4 uses the change of variable analytically and easier to implement.
>
> Therefore, we hope the reviewer can recognize the importance of Proposition 3.

---

> > ### Author Response · Authors · 2022-11-17
> > **Response2**
> >
> > ********Q: Originality and experiments. In the experimental setting, I think several more recent training-free baselines (e.g. [3,4]) are missing. I noticed that the authors discussed their relationship with the proposed methods in the appendix. I would prefer to discuss these relevant baselines and show quantitative comparison results in the main text.********
> >
> > A:
> >
> > 1.  ************Originality.************ In fact, DEIS appears on arXiv around **2 months easier** than DPM-Solver [4] (first available on Jun 2nd), Elucidating Diffusion Model [3]~(first available on Jun 1st and acknowledged DEIS). Therefore, we claim [3][4] and DEIS are concurrent work if DEIS is not an “existing” method for [3][4].
> > 2. We thank the discussions in openreview and comments from the DPM-Solver authors. Though the two algorithms share a lot of similarities, we find they have slightly different update schemes. We have made corrections accordingly in the revision and included more comparisons with DPM-Solver quantitatively and qualitatively. We find $t$AB-DEIS outperforms DPM-Solver in general.
> > 3. The second order Heun method used in Elucidating Diffusion Model is exactly $\rho$2Heun-DEIS in our work. We **have included** it in our empirical comparison (Tab 2). We find 3order Kutta methods outperform 2Heun when we have a relatively large NFE budget, and $t$AB-DEIS outperforms $\rho$RK-DEIS in CIFAR10, ImageNet and LSUN bedroom.
> > 4. We have added more discussions (Appendix B) and experiments in the revision to compare these methods. We feel it is more proper to have these comparisons in the appendix rather than the main text since they are concurrent work.
> >
> > ********Q: I suggest similar to many previous works (e.g., [4,5]) that consider ImageNet with a resolution of at least 64x64 for obtaining both quantitative and qualitative results.********
> >
> > A: We appreciate the reviewer’s suggestion. We have added experiments on ImageNet 64x64 and LSUN 256x256 with the pre-trained model from OpenAI. It shows DEIS outperforms existing methods. We have added ImageNet 256x256 for qualitative results.
> >
> > **************Q: In $p_{0t}(x_t | x_0)$ , what is $\mu_t$ ?**
> >
> > A: It is a matrix so that $\mu_t x_0$ is the mean of the Gaussian distribution $p_{0t}(x_t | x_0)$.
> >
> > ********Q: I’m not exactly sure what “+optimizing timesteps” in Figure 5 means. It seems that the main text doesn’t directly explain its meaning.********
> >
> > A: We find different timesteps schedule has some effects on FID. We optimize timesteps instead of using timesteps schedule suggested by the original works. We clean the figure and avoid distracting the reader’s attention.
> >
> > ********Q: we can present the idea with the noise prediction reparameterization, which has been commonly used in most diffusion models for images. The analysis for different network reparameterizations can be deferred to the appendix.********
> >
> > A: Though we agree with the reviewer’s other suggestions on writing, we tend to think this part is important for the analysis.
> >
> > 1. The ODE formulation is first available in [6] for continuous-time diffusion models and the score parameterization in ODE is more common in this setting.
> > 2. The comparison between the score parameterization and the noise prediction parameterization reveals that the parameterizations play an important role in controlling discretization errors. Though we can use advanced ODE solvers, such as RK method and exponential integrator, different parameterizations, like $\frac{dx}{dt} = F_t x - \frac{1}{2}G_t G^T_t s_\theta(x, t)$, $\frac{dx}{dt} = F_t x - \frac{1}{2}G_t G^T_t L_t^{-1}s_\theta(x, t)$ and $\frac{dy}{d\rho} = \epsilon_{\theta}(y, \rho)$ give quite different performance for a given ODE solver.
> > 3. The discussion also inspired us to find more accurate approximations for the behavior of networks in diffusion models, leading us to explore the choice of polynomial approximation.
> >
> > [1] J. Song et al. Denoising Diffusion Implicit Models
> >
> > [2] H. Lee et al. Convergence of Score-based Generative Modeling for General Data Distributions
> >
> > [3] T. Karras et al. Elucidating the Design Space of Diffusion-Based Generative Models
> >
> > [4] C. Lu et al. DPM-Solver: A Fast ODE Solver for Diffusion Probabilistic Model Sampling in Around 10 Steps
> >
> > [5] T. Karras et al. Progressive Distillation for Fast Sampling of Diffusion Models
> >
> > [6] Y. Song et al. Score-Based Generative Modeling through Stochastic Differential Equations

---

> > > ### Comment · Reviewer_xzme · 2022-11-23
> > > **Thank you for your response**
> > >
> > > The rebuttal has addressed my major concerns regarding the presentation and experimental results. In particular, regarding the importance of Proposition 3, I suggest the authors incorporate the above-mentioned many advantages, such as "it can apply any out-of-shelf ODE solver" and "easier to implement", etc to the main text to better motivate why you are doing so. Given the revised paper, I increased my rating from 5 to 6.

---

### Official Review · Reviewer_5JHW · 2022-11-02

**Confidence:** 2
**Correctness:** 3
**Technical Novelty And Significance:** 2
**Empirical Novelty And Significance:** 2
**Recommendation:** 5

**Clarity, Quality, Novelty And Reproducibility:**

The paper seems notationally and mathematically overcomplicated, idea is novel. I did not see link to the code whilst the method seems highly hard to replicate just from description as requires accurate invocation of ODE solvers.

**Strength And Weaknesses:**

Strengths:
1. Idea somewhat interesting from theoretical stand point
2. Detailed motivation for ODE case why this should work

Weakness:
1. It's not clear why you mention application to SDE case as you mention that your method do not work for integrating SDE equation
2. Notation is somewhat overcomplicated for no reason, e.g. \Phi(s, t) -- why you need to mention hard matrix ODE as in your case it is always scalar multiplied by identity matrix?

**Summary Of The Paper:**

Authors present efficient integrator for diffusion models based on polynomial interpolation of the noise during trajectory estimation.

**Summary Of The Review:**

The paper should be simplified and made more clear, at this point it is too hard to read due to unnecessary mathematical overload and the reproducibility is under question.

---

> ### Author Response · Authors · 2022-11-17
> **Response**
>
> Thank you for thoughtful feedback. We have updated our manuscript. We encourage the reviewer to take a look at our revision.
>
> ******Q: Contribution and summary of paper******
>
> A: The main contributions of our work include i) a principled study on the causes of errors in sampling from a diffusion model using numerical schemes, and ii) a novel numerical method that reduces discretization error and generates high quality samples with a small number of steps. Our DEIS method is a nontrivial synergy of many techniques. Beside polynomial interpolation, we have also integrated better reparametrization of ODE, exponential integrator, Runge-Kutta methods in solving probability flow ODE. All of these together lead to the DEIS algorithm that achieves the SOTA performance.
>
> ********************************Q: Reproducibility********************************
>
> A: We have added a clean implementation based on pytorch / jax and demo code in the supplementary material (we forgot to do so in the submission but have made our code available online before the submission). It is easy to reproduce our results claimed in the paper using this code.
>
> ********Q: It's not clear why you mention application to SDE case as you mention that your method do not work for integrating SDE equation********
>
> A: The exponential integrator (EI) is still applicable to SDE and **achieves better performance** than SDE samplers without EI, however, the performance is still not as good as that of DEIS for ODE. This observation is consistent with existing literatures on diffusion model sampling where ODE based methods normally have better performances [1]. This observation however does not exclude the possibility that SDE methods can eventually be better than ODE methods for diffusion model sampling. In fact, all the existing theoretical studies investigating the sampling complexity of diffusion models are all based on SDE [2].
>
> In this paper, we also establish some theoretical results on SDE based approaches and provide some explanations on why these methods could have worse performance when NFE is small. We believe these explanations and ``negative results'' are important steps to develop even better algorithms and are beneficial to the community. However, we have now moved the discussion (Previously Section 5) on SDE to the appendix to create more space for experiment results.
>
> ********Q: why you need to mention hard matrix ODE as in your case it is always scalar multiplied by identity matrix?********
>
> A: The DEIS method we develop is applicable to general diffusion models, including the critically-damped Langevin diffusion (CLD) **[3], and the Blurring diffusion models (BDM) [4] where matrix ODE is necessary. We use matrix ODE to keep the generality of our method. We have added a sentence  following Eq (1) to clarify this point.
>
> [1] J. Song et al. Denoising Diffusion Implicit Models
>
> [2] H. Lee et al. Convergence of Score-based Generative Modeling for General Data Distributions
>
> [3] T. Dockhorn et al. Score-Based Generative Modeling with Critically-Damped Langevin Diffusion
>
> [4] E. Hoogeboom et al. Blurring Diffusion Models

---

### Public Comment · ~Cheng_Lu5 · 2022-11-09
**Thank you for comparing with DPM-Solver!**

I highly appreciate it that the authors discuss the comparison between DPM-Solver and DEIS in Appendix B. These two works are concurrent and coincidently use the same technique called "exponential integrators" for accelerating the sampling by diffusion models.

However, despite the similarity, I think these two works have many differences, and some of the claims in Appendix B about DPM-Solver seem inaccurate. It would be much perfect if the authors could clarify the comparisons.

Below I list some of my concerns. As I did not fully understand this paper, it may have some mistakes. I would appreciate it if the authors could address the following questions and discuss the differences more.

## 1. DPM-Solver is not the special case of $\rho$RK-DEIS.
The $\rho$RK-DEIS uses a change-of-variable for time from $t$ to $\rho$, and for data from $x_t$ to $y_t$. Such change-of-variable makes the diffusion ODE become a pure ODE w.r.t. the noise prediction model. Then the author uses the classical Runge-Kutta methods to solve such ODE and name it as $\rho$RK-DEIS.

However, DPM-Solver uses a change-of-variable from $t$ to the log-SNR $\lambda=\log\alpha_t - \log\sigma_t$. Such a change-of-variable method is **another key contribution** of DPM-Solver because:
- It provides a very simplified formulation of the exact solution of diffusion ODEs and makes the sampling by diffusion ODEs invariant to the noise schedule (please see Appendix A in DPM-Solver for details).
- Because of such change-of-variable, the coefficients in the Taylor expansion are all **analytical** and can be exactly computed. (the coefficients are something like $\int e^{-\lambda}\lambda^k d\lambda$ and can be computed by repeatedly applying $k$ times of integration-by-parts method (and we have provided the detailed coefficients in the Appendix).

Moreover, on CIFAR-10 with continuous-time diffusion models, the FID results of DPM-Solver are also quite different from the $\rho$RK-DEIS. In fact, **the results of DPM-Solver are much better than those of $\rho$RK-DEIS**.

Therefore, as $\rho$RK-DEIS uses a different change-of-variable, it is essentially different from DPM-Solver. And I think the detailed updating equations of $\rho$RK-DEIS are different from those of DPM-Solver. Therefore, I don't think DPM-Solver is a special case of DEIS.

## 2. DPM-Solver can also accelerate the log-likelihood evaluation
As far as I understand, DEIS accelerates the log-likelihood evaluation by only accelerating the computation of the samples. As DPM-Solver can also accelerate the sampling, I still believe that, in principle, DPM-Solver (and all of the other fast samplers) can further accelerate the log-likelihood evaluations.

==============

In conclusion, I think the contributions of DPM-Solver are more than just applying the exponential integrators. It would be great if the authors could discuss and compare more about the differences. Thank you!

---

> ### Author Response · Authors · 2022-11-17
> **Thanks for your discussions**
>
> We are happy that authors from DPM-Solver raised the question and discussion openly. We believe the discussion can help clarify the connections and differences between DEIS and DPM-Solver.
>
> ********Q: Claims in Appendix B about DPM-Solver seem inaccurate. DPM-Solver is not the special case of $\rho$RK-DEIS. DPM-Solver uses a change-of-variable from $t$ to the log-SNR $\lambda$********
>
> 1. Thanks for the clarification. The two algorithms share a lot of similarities. For example, $t$ → $\rho$ in DEIS is almost equivalent to $t$ → $\lambda$ in DPM-Solver. Let us verify the connection with VPSDE  in Proposition 3. With DEIS notation, $\rho(t) = \sqrt{\alpha_0} (\sqrt{\frac{1-\alpha_t }{\alpha_t} } - \sqrt{\frac{1-\alpha_0 }{\alpha_0} })$. And usually, VPSDE takes $\alpha_0=1$ and we reach $\rho(t) = \sqrt{\frac{1-\alpha_t }{\alpha_t} }$. In DPM-Solver, for $\lambda(t) = \log \sqrt{\frac{\alpha_t}{1-\alpha_t } } = -\log \rho$.
> 2. The two algorithms indeed have different update schemes. We have made modification accordingly in the revision and included more comparisons. We encourage interested readers to check out our discussion section (Appendix B) that compares the two methods.
>
> ********************Q: DPM-Solver can also accelerate the log-likelihood evaluation. As far as I understand, DEIS accelerates the log-likelihood evaluation by only accelerating the computation of the samples.********************
>
> A: We respectfully disagree with the point. As admitted by DPM-Solver
>
> > DPM-Solver is designed for fast sampling, which may be not suitable for accelerating the likelihood evaluations of DPMs
> >
>
> It is unclear whether DPM solver has the same convergence order for evaluating log-likelihood similar as the order for sampling; it is non-trivial to accelerate augmented ODE in FFJORD, which is a popular tool to evaluate log-likelihood, with similar techniques developed in the DPM-Solver paper.
>
> On the other hand, we can apply existing out-of-box ODE solver and log-likelihood evaluation methods with the help of Proposition 3, which has absorbed analytical knowledge of diffusion coefficients and does not suffer from semi-linear stiff properties compared with direct working on the non-transformed ODE. Under proper assumptions,  we can show that the convergence order with transformed ODE for log-likelihood evaluation is the same as solving sampling ODE.
>
>
> We are happy for further discussions if readers have other questions or correct us if there are some mistakes.
>
> Thanks,

---

### Comment · Area_Chair_NMB2 · 2022-11-15
**Response**

Dear authors,

Your response to the reviews and public comment would be highly appreciated.

Kind regards,
Your AC

---

### Decision · Program_Chairs · 2023-01-20

**Decision:**

Accept: poster

**Justification For Why Not Higher Score:**

See Weaknesses.

**Justification For Why Not Lower Score:**

The authors wrote a strong rebuttal and did a substantial amount of work improving the paper based on the reviewer comments, adding larger-scale experiments and improving their writing. The appendix of this paper is also very detailed. It's obvious the authors put in a lot of time.

**Metareview: Summary, Strengths And Weaknesses:**

Ratings: 5/6/6/6.
Confidences: 2/3/4/3.
Recommendation: Accept.

The article discusses a new method for speeding up the slow sampling procedure in Diffusion Models (DMs), called the Diffusion Exponential Integrator Sampler (DEIS). The DEIS method uses the Exponential Integrator for discretizing ordinary differential equations and leverages the semilinear structure of the learned diffusion process to reduce the discretization error. The proposed method can generate high-fidelity samples in as few as 10 steps and outperforms previous methods in terms of sample quality and speed.

Strengths: (1) the proposed idea is novel and well-motivated. (2) The work provides good insight into the understanding of DDIM from a numerical discretization perspective. (3) Experimental results show the effectiveness of the proposed method.

Weaknesses: (1) The presentation of the proposed method is not very clear (2) The experiment settings and results are confusing and do not include recent training-free baselines. (3) Minor things, such as a lack of clarity in certain descriptions and terminology.

The authors wrote a comprehensive rebuttal, and after some discussion, one of the reviewers raised his score from 5 to 6.

Overall, I do recommend to accept this paper.

**Note From Pc:**

if the above contains the word "oral" or "spotlight" please see: "oral" presentation means -> notable-top-5% and "spotlight" means -> notable-top-25%. As stated in our emails, we are disassociating presentation type from AC recommendations